# CAN A MISL FLY? ANALYSIS AND INGREDIENTS FOR MUTUAL INFORMATION SKILL LEARNING

**Chongyi Zheng**[*] **Jens Tuyls**[*], **Joanne Peng & Benjamin Eysenbach**
Department of Computer Science
Princeton University
{chongyiz, jtuyls}@cs.princeton.edu

## ABSTRACT

Self-supervised learning has the potential of lifting several of the key challenges in reinforcement learning today, such as exploration, representation learning, and reward design. Recent work (METRA (Park et al., 2024)) has effectively argued that moving away from mutual information and instead optimizing a certain Wasserstein distance is important for good performance. In this paper, we argue that the benefits seen in that paper can largely be explained within the existing framework of mutual information skill learning (MISL). Our analysis suggests a new MISL method (contrastive successor features) that retains the excellent performance of METRA with fewer moving parts, and highlights connections between skill learning, contrastive representation learning, and successor features. Finally, through careful ablation studies, we provide further insight into some of the key ingredients for both our method and METRA.[1]

## 1 INTRODUCTION

Self-supervised learning has had a large impact on areas of machine learning ranging from audio processing (Oord et al., 2016; 2018) or computer vision (Radford et al., 2021; Chen et al., 2020) to natural language processing (Devlin et al., 2019; Radford & Narasimhan, 2018; Radford et al., 2019; Brown, 2020). In the reinforcement learning (RL) domain, the "right" recipe to apply self-supervised learning is not yet clear. Several self-supervised methods for RL directly apply off-the-shelf methods from other domains such as masked autoencoding (Liu et al., 2022), but have achieved limited success so far. Other methods design self-supervised routines more specifically built for the RL setting (Burda et al., 2019; Pathak et al., 2017; Eysenbach et al., 2019; Sharma et al., 2020; Pong et al., 2020). We will focus on the skill learning methods, which aim to learn a set of diverse and distinguishable behaviors (skills) without an external reward function. This objective is typically formulated as maximizing the mutual information between skills and states (Gregor et al., 2016; Eysenbach et al., 2019), namely *mutual information skill learning* (MISL). However, some promising recent advances in skill learning methods build on other intuitions such as Lipschitz constraints (Park et al., 2022) or transition distances (Park et al., 2023). This paper focuses on determining whether the good performance of those recent methods can still be explained within the well-studied framework of mutual information maximization.

METRA (Park et al., 2024), one of the strongest prior skill learning methods, proposes maximizing the Wasserstein dependency measure between states and skills as an alternative to the idea of mutual information maximization. The success of this method calls into question the viability of the MISL framework. However, mutual information has a long history dating back to Shannon (1948) and gracefully handles stochasticity and continuous states (Myers et al., 2024). These appealing properties of mutual information raises the question: *Can we build effective skill learning algorithms within the MISL framework, or is MISL fundamentally flawed?*

We start by carefully studying the components of METRA both theoretically and empirically. For representation learning, METRA maximizes a lower bound on the mutual information, resembling

---

[*]These authors contributed equally.
[1]Website and code: https://princeton-rl.github.io/contrastive-successor-features

contrastive learning. For policy learning, METRA optimizes a mutual information term *plus an extra exploration term*. These findings provide an interpretation of METRA that does not appeal to Wassertein distances and motivate a simpler algorithm (Fig. 1).

Building upon our new interpretations of METRA, we propose a simpler and competitive MISL algorithm called Contrastive Successor Features (CSF). First, CSF learns state representations by directly optimizing a contrastive lower bound on mutual information, preventing the dual gradient descent procedure adopted by METRA. Second, while any off-the-shelf RL algorithm (e.g. SAC (Haarnoja et al., 2018)) is applicable, CSF instead learns a policy by leveraging successor features of linear rewards defined by the learned representations. Experiments on six continuous control tasks show that CSF is comparable with METRA, as evaluated on exploration performance and on downstream tasks. Furthermore, ablation studies suggest that rewards derived from the information bottleneck as well as a specific parameterization of representations are key for good performance.

> **Key Takeaways**
> 1. METRA can be explained within the MISL framework: learning representations through maximizing MI (Sec. 4.1), and learning policies through maximizing MI *plus an exploration bonus* (Sec. 4.2).
> 2. Based on our understanding of METRA, we propose CSF (Sec. 5), a simple MISL algorithm that retains SOTA performance (Sec. 6.4).
> 3. We find several ingredients that are key to boost MISL performance (Sec. 6.3).

## 2 RELATED WORK

Through careful theoretical and experimental analysis, we develop a new mutual information skill learning method that builds upon contrastive learning and successor features.

**Unsupervised skill discovery.** Our work builds upon prior methods that perform unsupervised skill discovery. Prior work has achieved this aim by maximizing lower bounds (Tschannen et al., 2020; Poole et al., 2019) of different mutual information formulations, including diverse and distinguishable skill-conditioned trajectories (Li et al., 2023; Eysenbach et al., 2019; Hansen et al., 2020; Laskin et al., 2022; Strouse et al., 2022), intrinsic empowerment (Mohamed & Jimenez Rezende, 2015; Choi et al., 2021), distinguishable termination states (Gregor et al., 2016; Warde-Farley et al., 2019; Baumli et al., 2021), entropy bonus (Florensa et al., 2016; Lee

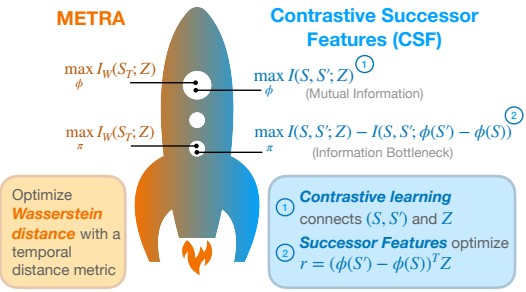

Figure 1: **From METRA to MISL. (Left)** METRA argues optimizing a Wasserstein distance is superior to using mutual information. **(Right)** Through careful analysis, we show METRA still bears striking similarities to MISL algorithms, which allows us to develop a new MISL algorithm (CSF) that matches the performance of METRA while retaining the theoretical properties associated with MI maximization.

et al., 2019; Shafiullah & Pinto, 2022), predictable transitions (Sharma et al., 2020; Campos et al., 2020), etc. Among those prior methods, perhaps the most related works are CIC (Laskin et al., 2022) and VISR (Hansen et al., 2020). We will discuss the difference between them and our method in Sec. 5. Another line of unsupervised skill learning methods utilize ideas other than mutual information maximization, such as Lipschitz constraints (Park et al., 2022), MDP abstraction (Park & Levine, 2023), model learning (Park et al., 2023), and Wasserstein distance (Park et al., 2024; He et al., 2022). Our work will analyze the state-of-the-art method named METRA (Park et al., 2024) that builds on the Wasserstein dependency measure (Ozair et al., 2019), provide an alternative explanation under the well-studied MISL framework, and ultimately develop a simpler method.

**Contrastive learning.** Contrastive learning has achieved great success for representation learning in natural language processing and computer vision (Radford et al., 2021; Chen et al., 2020; Gao et al., 2021; Sohn, 2016; Chopra et al., 2005; Oord et al., 2018; Gutmann & Hyvärinen, 2010; Ma & Collins, 2018; Tschannen et al., 2020). These methods aim to push together the representations of positive

pairs drawn from the joint distribution, while pushing away the representations of negative pairs drawn from the marginals (Ma & Collins, 2018; Oord et al., 2018). In the domain of RL, contrastive learning has been used to define auxiliary representation learning objective for control (Laskin et al., 2020; Yarats et al., 2021), solve goal-conditioned RL problems (Zheng et al., 2024; Eysenbach et al., 2022; 2020; Ma et al., 2023), and derive representations for skill discovery (Laskin et al., 2022). Prior work has also provided theoretical analysis for these methods from the perspective of mutual information maximization (Poole et al., 2019; Tschannen et al., 2020) and the geometry of learned representations (Wang & Isola, 2020). Our work will combine insights from both angles to analyze METRA and show its relationship to contrastive learning, resulting in a new skill learning method.

**Successor features.** Our work builds on successor representations (Dayan, 1993), which encode the discounted state occupancy measure of policies. Prior work has shown these representations can be learned on high-dimensional tasks (Kulkarni et al., 2016; Zhang et al., 2017) and help with transfer learning (Barreto et al., 2017). When combined with universal value function approximators (Schaul et al., 2015), these representations generalize to *universal* successor features, which estimates a value function for any reward under any policy (Borsa et al., 2019). While prior methods have combined successor feature learning with mutual information skill discovery for fast task inference (Hansen et al., 2020; Liu & Abbeel, 2021), we instead use successor features to replace $Q$ estimation after learning state representations (Sec. 5).

## 3 PRELIMINARIES

**Mutual information skill learning.** The MISL problem typically involves two steps: *(1)* unsupervised pretraining and *(2)* downstream control. For the first step, we consider a Markov decision process (MDP) *without* reward function defined by states $s \in \mathcal{S}$, actions $a \in \mathcal{A}$, initial state distribution $p_0 \in \Delta(\mathcal{S})$, discount factor $\gamma \in (0, 1]$, and dynamics $p : \mathcal{S} \times \mathcal{A} \mapsto \Delta(\mathcal{S})$, where $\Delta(\cdot)$ denotes the probability simplex. The goal of unsupervised pretraining is to learn a skill-conditioned policy $\pi : \mathcal{S} \times \mathcal{Z} \mapsto \Delta(\mathcal{A})$ that conducts diverse and discriminable behaviors, where $\mathcal{Z}$ is a latent skill space. We use $\beta : \mathcal{S} \times \mathcal{Z} \mapsto \Delta(\mathcal{A})$ to denote the behavioral policy. We select the prior distribution of skills as a uniform distribution over the $d$-dimensional unit hypersphere $p(z) = \mathrm{UNIF}(\mathbb{S}^{d-1})$ (a uniform von Mises–Fisher distribution (Wikipedia, 2024)) and will use this prior throughout our discussions to unify the theoretical analysis.

Given a latent skill space $\mathcal{Z}$, prior methods (Eysenbach et al., 2019; Sharma et al., 2020; Laskin et al., 2022; Gregor et al., 2016; Hansen et al., 2020) maximizes the MI between skills and states $I^{\pi}(S; Z)$ or the MI between skills and transitions $I^{\pi}(S, S'; Z)$ under the target policy. We will focus on $I^{\pi}(S, S'; Z)$ but our discussion generalizes to $I^{\pi}(S; Z)$. Specifically, maximizing the MI between skills and transitions can be written as

$$\max_{\pi} I^{\pi}(S, S'; Z) \overset{\text{const.}}{=} \max_{\pi} \mathbb{E}_{\substack{z \sim p(z), s \sim p^{\pi}(s_+ = s|z) \\ s' \sim p^{\pi}(s'|s,z)}} [\log p^{\pi}(z \mid s, s')], \tag{1}$$

where $p^{\pi}(s_+ = s \mid z)$ is the discounted state occupancy measure (Ho & Ermon, 2016; Nachum et al., 2019; Eysenbach et al., 2022; Zheng et al., 2024) of policy $\pi$ conditioned on skill $z$, and $p^{\pi}(s' \mid s, z)$ is the state transition probability given policy $\pi$ and skill $z$. This optimization problem can be casted into an iterative min-max optimization problem by first choosing a variational distribution $q(z \mid s, s')$ to fit the historical posterior $p^{\beta}(z \mid s, s')$, which is an approximation of $p^{\pi}(z \mid s, s')$, and then choosing policy $\pi$ to maximize discounted return defined by the intrinsic reward $\log q(z \mid s, s')$:

$$q_{k+1} \leftarrow \arg\max_{q} \mathbb{E}_{p^{\beta}(s,s',z)} [\log q(z \mid s, s')]. \tag{2}$$

$$\pi_{k+1} \leftarrow \arg\max_{\pi} \mathbb{E}_{p^{\pi}(s,s',z)} [\log q_k(z \mid s, s')], \tag{3}$$

where $k$ indicates the number of updates. See Appendix A.1 for detailed discussion.

For the second step, given a regular MDP (*with* reward function), we reuse the skill-conditioned policy $\pi$ to solve a downstream task. Prior methods achieved this aim by *(1)* reaching goals in a zero-shot manner (Park et al., 2022; 2023; 2024), *(2)* learning a hierarchical policy $\pi_h : \mathcal{S} \mapsto \Delta(\mathcal{Z})$ that outputs skills instead of actions (Eysenbach et al., 2019; Laskin et al., 2022; Gregor et al., 2016), or *(3)* planning in the latent space with a learned dynamics model (Sharma et al., 2020).

**METRA.** Maximizing the mutual information between states and latent skills $I(S; Z)$ only encourages an agent to find discriminable skills, while the algorithm might fail to prioritize state space coverage (Park et al., 2024; 2022). A prior state-of-the-art method METRA (Park et al., 2024) proposes to solve this problem by learning representations of states $\phi : \mathcal{S} \mapsto \mathbb{R}^d$ via maximizing the Wasserstein dependency measure (WDM) (Ozair et al., 2019) between states and skills $I_{\mathcal{W}}(S; Z)$. Specifically, METRA chooses to enforce the 1-Lipschitz continuity of $\phi$ under the temporal distance metric, resulting in a constrained optimization problem for $\phi$:

$$\max_{\phi} \mathbb{E}_{p(z)p^{\beta}(s,s'|z)}[(\phi(s') - \phi(s))^{\top} z] \quad \text{s.t. } \|\phi(s') - \phi(s)\|_2^2 \leq 1 \; \forall (s, s') \in \mathcal{S}_{\text{adj}}^{\beta}, \quad (4)$$

where $p^{\beta}(s, s' \mid z)$ denotes the probability of first sampling $s$ from the discounted state occupancy measure $p^{\beta}(s_+ = s \mid z)$ and then transiting to $s'$ by following the behavioral policy $\beta$, and $\mathcal{S}_{\text{adj}}^{\beta}$ denotes the set of all the adjacent state pairs visited by $\beta$. In practice, METRA uses dual gradient descent to solve Eq. 4, resulting in an iterative optimization problem[2]

$$\min_{\lambda \geq 0} \max_{\phi} \; L(\phi, \lambda)$$

$$L(\phi, \lambda) \triangleq \mathbb{E}_{p(z)p^{\beta}(s,s'|z)}[(\phi(s') - \phi(s))^{\top} z] + \lambda \left(1 - \mathbb{E}_{p^{\beta}(s,s')} \left[\|\phi(s') - \phi(s)\|_2^2\right]\right), \quad (5)$$

Importantly, $L(\phi, \lambda)$ is *not* the Lagrangian of Eq. 4 because $L(\phi, \lambda)$ does not contain a dual variable for every $(s, s') \in \mathcal{S}_{\text{adj}}^{\beta}$. We will discuss the actual METRA representation objective and the behavior of convergent representations in Sec. 4.1.

After learning the state representation $\phi$, METRA finds its skill-conditioned policy $\pi$ via maximizing the RL objective with intrinsic reward $(\phi(s') - \phi(s))^{\top} z$:

$$\max_{\pi} J(\pi), J(\pi) \triangleq \mathbb{E}_{\substack{z \sim p(z), s \sim p^{\pi}(s_+ = s|z) \\ s' \sim p^{\pi}(s'|s,z)}} \left[(\phi(s') - \phi(s))^{\top} z\right]. \quad (6)$$

In the following sections, we will provide another way to understand this SOTA method, draw connections with contrastive learning (Oord et al., 2018) and the information bottleneck (Alemi et al., 2017), and then derive a simpler MISL algorithm.

## 4 UNDERSTANDING THE PRIOR METHOD

In this section we reinterpret METRA through the lens of MISL, showing that:

1. The METRA representation objective is nearly identical to a contrastive loss (which maximizes a lower bound on mutual information). See Sec. 4.1.

2. The METRA actor objective is equivalent to a mutual information lower bound *plus an extra term*. This extra term is related to an information bottleneck (Tishby et al., 2000; Alemi et al., 2017) and our experiments will show it is important for exploration. See Sec. 4.2.

Sec. 5 will then introduce a new mutual information algorithm that combines these insights to match the performance of METRA while *(1)* retaining the theoretical grounding of mutual information and *(2)* being simpler to implement.

### 4.1 CONNECTING METRA'S REPRESENTATION OBJECTIVE AND CONTRASTIVE LEARNING

Our understanding of METRA starts by interpreting the representation objective of METRA as a contrastive loss. This interpretation proceeds by two steps. First, we focus on understanding the *actual* representation objective of METRA, aiming to predict the convergent behavior of the learned representations. Second, based on the actual representation objective, we draw a connection between METRA and contrastive learning. In Sec. 6, we conduct experiments to verify that METRA learns optimal representations in practice and that they bear resemblance to contrastive representations.

Sec. 3 mentioned that the Lagrangian $L(\phi, \lambda)$ used as the METRA representation objective does not correspond to the constrained optimization problem in Eq. 4, raising the following question: *What is the actual METRA representation objective?* To answer this question, we note that, rather than using distinct dual variables for each pair of $(s, s') \in \mathcal{S}_{\text{adj}}^{\beta}$, $L(\phi, \lambda)$ employs a single dual variable, imposing

---

[2]We ignore the slack variable $\epsilon$ in Park et al. (2024) because it takes a fairly small value $\epsilon = 10^{-3} \ll 1$.

an *expected* temporal distance constraint over all pairs of $(s, s')$ under the historical transition distribution $p^\beta(s, s')$. This observation suggests that METRA's representations are optimized with the following objective

$$\max_\phi \mathbb{E}_{p(z)p^\beta(s,s'|z)}[(\phi(s') - \phi(s))^\top z] \quad \text{s.t. } \mathbb{E}_{p^\beta(s,s')} \left[\|\phi(s') - \phi(s)\|_2^2\right] \leq 1. \quad (7)$$

Applying KKT conditions to $L(\phi, \lambda)$, we claim that

**Proposition 1.** *The optimal state representation $\phi^\star$ of the actual METRA representation objective (Eq. 7) satisfies*

$$\mathbb{E}_{p^\beta(s,s')} \left[\|\phi^\star(s') - \phi^\star(s)\|_2^2\right] = 1.$$

The proof is in Appendix A.2. Constraining the representation of consecutive states in expectation not only clarifies the actual METRA representation objective, but also means that we can predict the value of this expectation for optimal $\phi$. Sec. 6.1 includes experiments studying whether the optimal representation satisfies this proposition in practice. Importantly, identifying the actual METRA representation objective allows us to draw a connection with the rank-based contrastive loss (InfoNCE (Oord et al., 2018; Ma & Collins, 2018)), which we discuss next.

We relate the actual METRA representation objective to a contrastive loss, which we will specify first and then provide some intuitions for what it is optimizing. This loss is a lower bound on the mutual information $I^\beta(S, S'; Z)$ and a variant of the InfoNCE objective (Henaff, 2020; Ma & Collins, 2018; Zheng et al., 2024). Starting from the standard variational lower bound (Barber & Agakov, 2004; Poole et al., 2019), prior work derived an unnormalized variational lower bound on $I^\beta(S, S'; Z)$ ($I_{\text{UBA}}$ in (Poole et al., 2019)),

$$I^\beta(S, S'; Z) \geq \mathbb{E}_{p^\beta(s,s',z)}[f(s, s', z)] - \mathbb{E}_{p^\beta(s,s')} \left[\log \mathbb{E}_{p(z')} \left[e^{f(s,s',z')}\right]\right],$$

where $f : \mathcal{S} \times \mathcal{S} \times \mathcal{Z} \mapsto \mathbb{R}$ is the critic function (Ma & Collins, 2018; Poole et al., 2019; Eysenbach et al., 2022; Zheng et al., 2024). Since the critic function $f$ takes arbitrary functional form, one can choose to parameterize $f$ as the inner product between the difference of transition representations and the latent skill, i.e. $f(s, s', z) = (\phi(s') - \phi(s))^\top z$. This yields a specific lower bound:

$$I^\beta(S, S'; Z) \geq \underbrace{\mathbb{E}_{p^\beta(s,s',z)}[(\phi(s') - \phi(s))^\top z]}_{\text{LB}_+^\beta(\phi)} - \underbrace{\mathbb{E}_{p^\beta(s,s')} \left[\log \mathbb{E}_{p(z')} \left[e^{(\phi(s') - \phi(s))^\top z'}\right]\right]}_{\text{LB}_-^\beta(\phi)} \triangleq \text{LB}^\beta(\phi). \quad (8)$$

Intuitively, $\text{LB}_+^\beta(\phi)$ pushes together the difference of transition representations $\phi(s') - \phi(s)$ and the latent skill $z$ sampled from the same trajectory (positive pairs), while $\text{LB}_-^\beta(\phi)$ pushes away $\phi(s') - \phi(s)$ and $z$ sampled from different trajectories (negative pairs). This intuition is similar to the effects of the contrastive loss, and we note that Eq. 8 only differs from the standard InfoNCE loss in excluding the positive pair in $\text{LB}_-^\beta(\phi)$. We will call this lower bound on the mutual information the *contrastive lower bound*.

We now connect the contrastive lower bound $\text{LB}^\beta(\phi)$ (Eq. 8) to the actual METRA representation loss $L(\phi, \lambda)$ (Eq. 5). While both of these optimization problems share the positive pair term ($\text{LB}_+^\beta(\phi)$), they vary in the way they handle randomly sampled $(s, s', z)$ pairs (negatives): METRA constrains the expected L2 representation distances $\lambda \left(1 - \mathbb{E}_{p^\beta(s,s')} \left[\|\phi(s') - \phi(s)\|_2^2\right]\right)$, while the contrastive lower bound minimizes the log-expected-exp score ($\text{LB}_-^\beta(\phi)$). However, we bridge this difference by viewing the expected L2 distance as a quadratic approximation of the log-expected-exp score:

**Proposition 2.** *There exists a $\lambda_0(d)$ depending on the dimension $d$ of the state representation $\phi$ such that the following second-order Taylor approximation holds*

$$\lambda_0(d)(1 - \mathbb{E}_{p^\beta} \left[\|\phi(s') - \phi(s)\|_2^2\right]) \approx LB_-^\beta(\phi).$$

See Appendix A.3 for a proof. This approximation shows that the constraint in the actual METRA representation loss has effects similar to $\text{LB}_-^\beta(\phi)$, namely pushing $\phi(s') - \phi(s)$ away from randomly sampled skills. Furthermore, this proposition allows us to spell out the (approximate) equivalence between representation learning in METRA and the contrastive lower bound on $I^\beta(S, S'; Z)$:

**Corollary 1.** *The METRA representation objective is equivalent to a second-order Taylor approximation of $I^\beta(S, S'; Z)$, i.e., $L(\phi, \lambda_0(d)) \approx LB^\beta(\phi)$.*

The METRA representation objective can be interpreted as a contrastive loss, allowing us to predict that the optimal state representations $\phi^\star$ (Prop. 1) have properties similar to those learned via contrastive learning. In Appendix C.1, we include experiments studying whether the approximation in Prop. 2 is reasonable in practice. In Sec. 6.2, we empirically compare METRA's representations to those learned by the contrastive loss. Sec. 6.3 will study whether replacing the METRA representation objective with a contrastive objective retains similar performance.

## 4.2    CONNECTING METRA'S ACTOR OBJECTIVE WITH AN INFORMATION BOTTLENECK

This section discusses the actor objective used in METRA. We first clarify the distinction between the actor objective of METRA and those used in prior methods, helping to identify a term that discourages exploration. Removing this anti-exploration term results in covering a larger proposition of the state space while learning distinguishable skills. We then relate this anti-exploration term to estimating another mutual information, drawing a connection between the entire METRA actor objective and a variant of the information bottleneck (Tishby et al., 2000; Alemi et al., 2017).

While prior work (Eysenbach et al., 2019; Gregor et al., 2016; Sharma et al., 2020; Hansen et al., 2020; Campos et al., 2020) usually uses the same functional form of the lower bound on (Eq. 2 & 3) different variants of the mutual information to learn both representations and skill-conditioned policies (see Appendix A.5 for details), METRA uses different objectives for the representation and the actor. Specifically, the actor objective of METRA $J(\pi)$ (Eq. 6) only encourages the similarity between the difference of transition representations $\phi(s') - \phi(s)$ and their skill $z$ (positive pairs), while ignoring the dissimilarity between $\phi(s') - \phi(s)$ and a random skill $z$ (negative pairs):

$$J(\pi) = \mathrm{LB}^\pi_+(\phi) = \mathrm{LB}^\pi(\phi) - \mathrm{LB}^\pi_-(\phi),$$

where $\mathrm{LB}^\pi(\phi)$, $\mathrm{LB}^\pi_+(\phi)$, and $\mathrm{LB}^\pi_-(\phi)$ are under the target policy $\pi$ instead of the behavioral policy $\beta$. The SOTA performance of METRA and the divergence between the functional form of the actor objective (positive term) and the representation objective (positive and negative terms) suggests that $\mathrm{LB}^\pi_-(\phi)$ may be a term discouraging exploration. Intuitively, removing this anti-exploration term boosts the learning of diverse skills. We will empirically study the effect of the anti-exploration term in Sec. 6.3 and provide theoretical interpretations next.

Our understanding of the anti-exploration term $\mathrm{LB}^\pi_-(\phi)$ relates it to a resubstituion estimation of the differential entropy $h^\pi(\phi(S') - \phi(S))$ in the representation space (see Appendix A.4 for details), i.e., $\mathrm{LB}^\pi_-(\phi) = \hat{h}^\pi(\phi(S') - \phi(S))$. Note that this entropy is different from the entropy of states $h^\pi(S)$, indicating that we want to *minimize* the entropy of difference of representations $\phi(s') - \phi(s)$ to encourage exploration. There are two underlying reasons for this (seemly counterintuitive) purpose: METRA aims to *(1)* constrain the expected L2 distance of difference of representations $\phi(s') - \phi(s)$ (Eq. 5) and *(2)* push difference of representations $\phi(s') - \phi(s)$ towards skills $z$ sampled from $\mathrm{U{\small NIF}}(\mathbb{S}^{d-1})$. Nonetheless, this relationship allows us to further rewrite the anti-exploration term $\mathrm{LB}^\pi_-(\phi)$ as an estimation of the mutual information $I^\pi(\phi(S') - \phi(S'); S, S')$, connecting the METRA actor objective to an information bottleneck:

**Proposition 3.** *The METRA actor objective is a lower bound on the information bottleneck $I^\pi(S, S'; Z) - I^\pi(S, S'; \phi(S') - \phi(S))$, i.e., $J(\pi) \leq I^\pi(S, S'; Z) - I^\pi(S, S'; \phi(S') - \phi(S))$.*

See Appendix A.4 for a proof and further discussions. Maximizing the information bottleneck $I^\pi(S, S'; Z) - I^\pi(S, S'; \phi(S') - \phi(S))$ compresses the information in transitions $(s, s')$ into difference in representations $\phi(s') - \phi(s)$ while relating these representations to the latent skills $z$ (Alemi et al., 2017; Tishby et al., 2000). This result implies that simply maximizing the mutual information $I^\pi(S, S'; Z)$ may be insufficient for deriving a diverse skill-conditioned policy $\pi$, and removing the anti-exploration $\mathrm{LB}^\pi_2(\phi)$ may be a key ingredient for the actor objective. In Appendix A.7, we propose a general MISL framework based on Prop. 3.

## 5    A SIMPLIFIED ALGORITHM FOR MISL VIA CONTRASTIVE LEARNING

In this section, we derive a simpler unsupervised skill learning method building upon our understanding of METRA (Sec. 4). This method maximizes MI (unlike METRA), while retaining the good

performance of METRA (see discussion in Sec. 3). We will first use the contrastive lower bound to optimize the state representation $\phi$ and estimate intrinsic rewards, and then we will learn the policy $\pi$ using successor features. We use **contrastive successor features (CSF)** to refer to our method.

## 5.1 Learning Representations through Contrastive Learning

Based on our analysis in Sec. 4.1, we use the contrastive lower bound on $I^\beta(S, S'; Z)$ to optimize the state representation directly. Unlike METRA, we obtain this contrastive lower bound *within* the MISL framework (Eq. 2 & 3) by employing a parameterization of the variational distribution $q(z \mid s, s')$ mentioned in prior work (Poole et al., 2019; Song & Kingma, 2021). Specifically, using a scaled energy-based model conditioned representations of transition pairs $(s, s')$, we define the variational distribution as

$$q(z \mid s, s') \triangleq \frac{p(z)e^{(\phi(s') - \phi(s))^\top z}}{\mathbb{E}_{p(z')}\left[e^{(\phi(s') - \phi(s))^\top z'}\right]}. \tag{9}$$

Plugging this parameterization into Eq. 2 produces

$$\phi_{k+1} \leftarrow \arg\max_\phi \mathbb{E}_{p^\beta(s, s', z)}\left[(\phi(s') - \phi(s))^\top z\right] - \mathbb{E}_{p^\beta(s, s')}\left[\log \mathbb{E}_{p(z')}\left[e^{(\phi(s') - \phi(s))^\top z'}\right]\right], \tag{10}$$

which is exactly the contrastive lower bound on $I^\beta(S, S'; Z)$. This contrastive lower bound allows us to learn the state representation $\phi$ while getting rid of the dual gradient descent procedure (Eq. 5) adopted by METRA. In practice, we find that adding a fixed coefficient $\xi = 5$ to the second term of Eq. 10 helps boost performance. We include further discussions of $\xi$ in Appendix A.6 and ablation studies in Appendix C.3.

In the same way that the METRA actor objective excluded the anti-exploration term (Sec. 4.2), we propose to construct the intrinsic reward by removing the negative term from our representation objective (Eq. 10), resulting in the same RL objective as $J(\pi)$ (Eq. 6):

$$\pi_{k+1} \leftarrow \arg\max_\pi \mathbb{E}_{p^\pi(s, s', z)}\left[r_k(s, s', z)\right], r_k(s, s', z) \triangleq (\phi_k(s') - \phi_k(s))^\top z \tag{11}$$

We use this RL objective as the update rule for the skill-conditioned policy $\pi$ in our algorithm.

## 5.2 Learning a Policy with Successor Features

To optimize the policy (Eq. 11), we will use an actor-critic method. Most skill learning methods use an off-the-shelf RL algorithm (e.g., TD3 (Fujimoto et al., 2018), SAC (Haarnoja et al., 2018)) to fit the critic. However, by noting that the intrinsic reward function $r(s, s', z)$ [3] is a linear combination between basis $\phi(s') - \phi(s) \in \mathbb{R}^d$ and weights $z \in \mathcal{Z} \subset \mathbb{R}^d$, we can borrow ideas from successor representations to learn a vector-valued critic. We learn the successor features $\psi^\pi : \mathcal{S} \times \mathcal{A} \times \mathcal{Z} \mapsto \mathbb{R}^d$:

$$\psi^\pi(s, a, z) \triangleq \mathbb{E}_{s \sim p^\pi(s_+ = s \mid z), s' \sim p(s' \mid s, a)}\left[\phi(s') - \phi(s)\right],$$

with the corresponding skill-conditioned policy $\pi$ in an actor-critic style:

$$\psi_{k+1}(s, a, z) \leftarrow \arg\min_\psi \mathbb{E}_{(s, a, z) \sim p^\beta(s, a, s', z), a' \sim \pi(a' \mid s', z)}\left[\left(\psi(s, a, z) - \hat{\psi}_k(s, s', a', z)\right)^2\right],$$

$$\text{where} \quad \hat{\psi}_k(s, s', a', z) \triangleq \phi_k(s') - \phi_k(s) + \gamma \bar{\psi}_k(s', a', z),$$

$$\pi_{k+1} \leftarrow \arg\max_\pi \mathbb{E}_{(s, z) \sim p^\beta(s, z), a \sim \pi(a \mid s, z)}\left[\psi_k(s, a, z)^\top z\right],$$

where $\psi$ is an estimation of $\psi^\pi$. In practice, we optimize $\psi$ and $\pi$ for one gradient step iteratively.

**Algorithm Summary.** In Alg. 1, we summarize CSF, our new algorithm.[4] Starting from an existing MISL algorithm (e.g., DIAYN (Eysenbach et al., 2019) and METRA (Park et al., 2024)), implementing our algorithm requires making three simple changes: *(1)* learning state representations $\phi_\theta$ by minimizing an InfoNCE loss (excluding positive pairs in the denominator) between pairs of

---

[3]We ignore the iteration $k$ for notation simplicity.
[4]Code: https://github.com/Princeton-RL/contrastive-successor-features

---

**Algorithm 1** Contrastive Successor Features

---

1: **Input** state representations $\phi_\theta$, successor features $\psi_\omega$, skill-conditioned policy $\pi_\eta$, and target successor feature $\psi_{\bar\omega}$.
2: **for** each iteration **do**
3:     Collect trajectory $\tau$ with $z \sim p(z)$ and $a \sim \pi_\eta(a \mid s, z)$, and then add $\tau$ to the replay buffer.
4:     Sample $\{(s, a, s', z)\} \sim$ replay buffer, $\{a'\} \sim \pi_\eta(a' \mid s', z)$, and $\{z'\} \sim p(z')$.
5:     $\mathcal{L}(\theta) \leftarrow -\mathbb{E}_{(s,s',z)} \left[ (\phi_\theta(s') - \phi_\theta(s))^\top z \right] + \mathbb{E}_{(s,s')} \left[ \log \sum_{z'} e^{(\phi_\theta(s') - \phi_\theta(s))^\top z'} \right]$.
6:     $\mathcal{L}(\omega) \leftarrow \mathbb{E}_{(s,a,s',a',z)} \left[ (\psi_\omega(s, a, z) - (\phi_\theta(s') - \phi_\theta(s) + \gamma\psi_{\bar\omega}(s', a', z)))^2 \right]$.
7:     $\mathcal{L}(\eta) \leftarrow -\mathbb{E}_{(s,z),a\sim\pi_\eta(a|s,z)} \left[ \psi_\omega(s, a, z)^\top z \right]$.
8:     Update $\theta$, $\omega$, and $\eta$ by taking gradients of $\mathcal{L}(\theta)$, $\mathcal{L}(\omega)$, and $\mathcal{L}(\eta)$.
9:     Update $\bar\omega$ using exponential moving averages.
10: **Return** $\phi_\theta$, $\psi_\omega$, and $\pi_\eta$.

---

$(s, s')$ and $z$, *(2)* using a critic $\psi_\omega$ with $d$-dimensional outputs and replacing the scalar reward with the vector $\phi_\theta(s') - \phi_\theta(s)$, *(3)* sampling the action $a$ from the policy $\phi_\eta$ to maximize the inner product $\psi_\omega(s, a, z)^\top z$.

Unlike CIC (Laskin et al., 2022), our method does not use the standard InfoNCE loss and instead employs a variant of it. Unlike VISR (Hansen et al., 2020), our method does not train the state representation $\phi$ using a skill discriminator. Unlike METRA, our method learns representations using the contrastive lower bound directly, avoids the Wasserstein distance and dual gradient descent optimization, and results in a simpler algorithm (see Appendix B.1 for further discussions).

## 6 EXPERIMENTS

The aims of our experiments are *(1)* verifying the theoretical analysis in Sec. 4 experimentally, *(2)* identifying several ingredients that are key to making MISL algorithms work well more broadly, and *(3)* comparing our simplified algorithm CSF to prior work. Our experiments will use standard benchmarks introduced by prior work on skill learning. All experiments show means and standard deviations across ten random seeds.

### 6.1 METRA CONSTRAINS REPRESENTATIONS IN EXPECTATION

Sec. 4.1 predicts that the optimal METRA representation satisfies its constraint $\mathbb{E}_p^\beta(s, s') \left[ \|\phi(s') - \phi(s)\|_2^2 \right] = 1$ strictly (Prop. 1). We study whether this condition holds after training the algorithm for a long time. To answer this question, we conduct didactic experiments with the state-based `Ant` from METRA (Park et al., 2024) navigating in an open space. We set the dimension of $\phi$ to $d = 2$ such that visualizing the learned representations becomes easier. After training the METRA algorithm for 20M environment steps (50K gradient steps), we analyze the norm of the difference in representations $\|\phi(s') - \phi(s)\|_2^2$.

We plot the histogram of $\|\phi(s') - \phi(s)\|_2^2$ over 10K transitions randomly sampled from the replay buffer (Fig. 2a). The observation that the empirical average of $\|\phi(s') - \phi(s)\|_2^2$ converges to 0.9884 suggests that the learned representations are feasible. Stochastic gradient descent methods typically find globally optimal solutions on over-parameterized neural networks (Du et al., 2019), making us conjecture that the learned representations are nearly optimal (Prop. 1). Furthermore, the spreading of the value of $\|\phi(s') - \phi(s)\|_2^2$ implies that maximizing the METRA representation objective will *not* learn state representations $\phi$ that satisfy $\|\phi(s') - \phi(s)\|_2^2 \leq 1$ for every $(s, s') \in \mathcal{S}_{\text{adj}}^\beta$. These results help to explain what objective METRA's representations are optimizing.

### 6.2 METRA LEARNS CONTRASTIVE REPRESENTATIONS

We next study connections between representations learned by METRA and those learned by contrastive learning empirically. Our analysis in Sec. 4.1 reveals that the representation objective of METRA corresponds to the contrastive lower bound on $I^\beta(S, S'; Z)$. This analysis raises the question whether representations learned by METRA share similar structures to representations learned by contrastive losses (Gutmann & Hyvärinen, 2010; Ma & Collins, 2018; Wang & Isola, 2020).

To answer this question, we reuse the trained algorithm in Sec. 6.1 and visualize two important statistics: *(1)* the conditional differences in representations $\phi(s') - \phi(s) - z$ and *(2)* the normalized marginal differences in representations $(\phi(s') - \phi(s))/\|\phi(s') - \phi(s)\|_2$. The resulting histograms (Fig. 2b & 2c) indicate that the conditional differences in representations $\phi(s') - \phi(s) - z$ converges to an isotropic Gaussian in distribution while the normalized marginal differences in representations

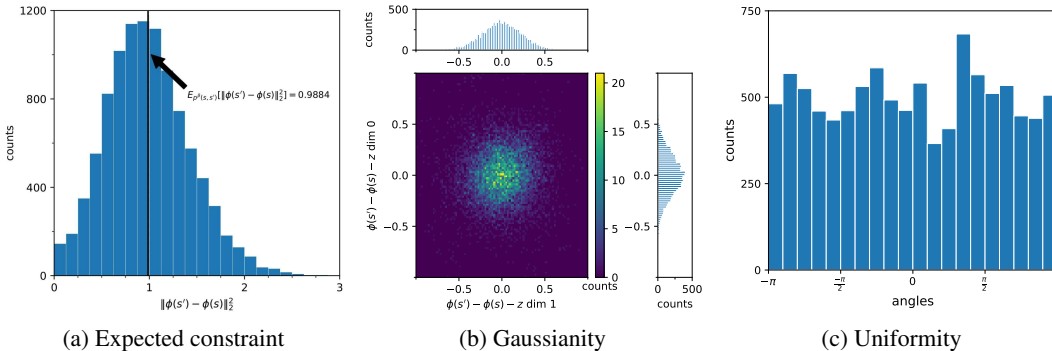

(a) Expected constraint        (b) Gaussianity        (c) Uniformity

Figure 2: **Histograms of METRA representations.** *(a)* The expected distance of representations converges to $1.0$, helping to explain what objective METRA's representations are optimizing. *(b)* Given a latent skill, the conditional difference in representations $(\phi(s') - \phi(s) \mid z)$ converges to an isotropic Gaussian distribution. *(c)* Taking the marginal over latent skills, the normalized difference in representations $\left( \frac{(\phi(s') - \phi(s))}{\|\phi(s') - \phi(s)\|_2} \right)$ converges to a $\mathrm{UNIF}(\mathbb{S}^{d-1})$. These observations are consistent with our theoretical analysis (Cor. 1) suggesting that METRA is performing a form of contrastive learning.

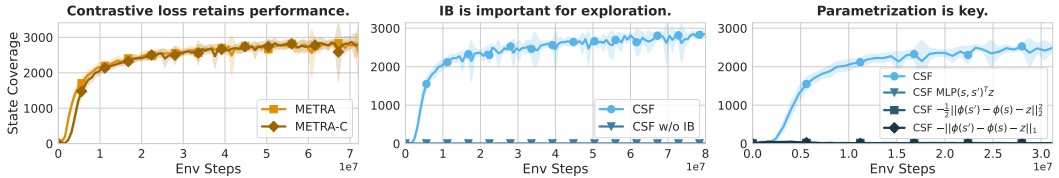

Figure 3: **Ablation studies.** *(Left)* Replacing the METRA representation loss with a contrastive loss retains performance. *(Center)* Using an information bottleneck to define the intrinsic reward is important for MISL. *(Right)* Choosing the right parameterization is crucial for good performance. Shaded areas indicate 1 std. dev.

$(\phi(s') - \phi(s))/\|\phi(s') - \phi(s)\|_2$ converges to a uniform distribution on the $d$-dimensional unit hypersphere $\mathbb{S}^{d-1}$ in distribution. Prior work (Wang & Isola, 2020) has shown that representations derived from contrastive learning preserves properties similar to these observations. We conjecture that maximizing the contrastive lower bound on $I^\beta(S, S'; Z)$ directly has the same effect as maximizing the METRA representation objective. See Appendix A.8 for formal claims and connections.

## 6.3 ABLATION STUDIES

We now study various design decisions of both METRA and CSF, aiming to identify some key factors that boost these MISL algorithms. We will conduct ablation studies on `Ant` again, comparing coverage of $(x, y)$ coordinates of different variants.

**(1) Contrastive learning recovers METRA's representation objective.** Our analysis (Sec. 4.1) and experiments (Sec. 6.2) have shown that METRA learns contrastive representations. We now test whether we can retain the performance of METRA by simply replacing its representation objective with the contrastive lower bound (Eq. 8). Results in Fig. 3 *(Left)* suggest that using the contrastive loss (METRA-C) fully recovers the original performance, circumventing the Wasserstein dependency measure.

**(2) Maximizing the information bottleneck is important.** In Sec. 4.2, we interpret the intrinsic reward in METRA as a lower bound on an information bottleneck. We conduct ablation experiments to study the effect of maximizing this information bottleneck over maximizing the mutual information directly, a strategy typically used by prior methods (Eysenbach et al., 2019; Mendonca et al., 2021; Hansen et al., 2020). Results in Fig. 3 *(Center)* show that CSF failed to discover skills when only maximizing the mutual information (i.e. including the anti-exploration term). These results indicate that using the information bottleneck as the intrinsic reward may be important for MISL algorithms.

**(3) Parameterization is key for CSF.** When optimizing a lower bound on the mutual information $I^\pi(S, S'; Z)$ using a variational distribution, there are many ways to parametrize the critic $f(s, s', z)$. In Eq. 9, we chose the parameterization $(\phi(s') - \phi(s))^\top z$, but there are many other choices. Testing the sensitivity of this choice of parameterization allows us to determine whether a *specific form* of the lower bound is important. In Fig. 3, we study several variants of CSF that use *(1)* a monolithic network $\mathrm{MLP}(s, s')^\top z$, *(2)* a Gaussian kernel $(-\frac{1}{2}\|\phi(s') - \phi(s)\|_2^2)$, or *(3)* a Laplacian kernel $(-\|\phi(s') - \phi(s)\|_1)$ as the critic parameterization. We find the alternative parameterizations are catastrophic for performance, suggesting that the inner product parameterization is key to CSF. We provide some insights for this parameterization in Appendix A.9.

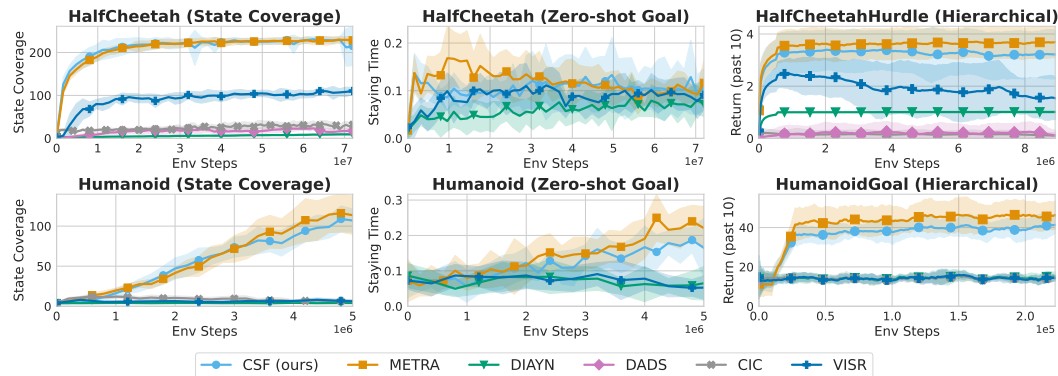

Figure 4: **CSF performs on par with METRA.** We compare CSF with baselines on state coverage *(left)*, zero-shot goal reaching *(middle)*, and hierarchical control *(right)*. CSF performs roughly on par with METRA and outperform all other baselines in most settings. Shaded areas indicate one standard deviation. Appendix Fig. 5, 6& 7 show the learning curves for all tasks.

## 6.4 CSF MATCHES SOTA FOR BOTH EXPLORATION AND DOWNSTREAM PERFORMANCE

Our final set of experiments compare CSF to prior MISL algorithms, measuring performance on both unsupervised exploration and solving downstream tasks.

**Experimental Setup.** We evaluate on the same five tasks as those used in Park et al. (2024) plus `Robobin` from LEXA (Mendonca et al., 2021). For baselines, we also use a subset from Park et al. (2024) (METRA (Park et al., 2024), CIC (Laskin et al., 2022), DIAYN (Eysenbach et al., 2019), and DADS (Sharma et al., 2020)) along with VISR (Hansen et al., 2020). See Appendix B.2 for details.

**Exploration performance.** To measure the unsupervised exploration capabilities of each method, we compute the state coverage by counting the unique number of $(x, y)$ coordinates visited by the agent. Fig. 4 *(left)* shows CSF matches METRA on both `HalfCheetah` and `Humanoid`. See Appendix B.3 for full results on exploration.

**Zero-shot goal reaching.** In this setting, the agent infers the right skill given a goal without further training on the environment. We evaluate on the same set of six tasks and defer both the goal sampling and skill inference strategies to Appendix B.4. We report the *staying time fraction*, which is the number of time steps that the agent stays at the goal divided by the horizon length. In Fig. 4 *(middle)*, we find all methods to perform similarly on `HalfCheetah`, while METRA and CSF perform best on `Humanoid`, with METRA performing slightly better on the latter. See Appendix B.4 for full results on zero-shot goal reaching.

**Hierarchical control.** We train a hierarchical controller $\pi_h(z \mid s)$ that outputs latent skills $z$ as actions for every fixed number of time steps to maximize the discounted return in two downstream tasks from Park et al. (2024), one of which requires to reach a specified goal (`HumanoidGoal`) and one requires jumping over hurdles (`HalfCheetahHurdle`). The results in Fig. 4 *(right)* show CSF and METRA are the best performing methods, showing mostly similar performance. See Appendix B.5 for full results and details.

Taken together, CSF is a competitive MISL algorithm that matches the current SOTA. On the full set of results (Appendices B.3, B.4, and B.5) we find that CSF continues to perform roughly on par with METRA on most tasks, though there are some tasks where CSF performs better and vice versa.

## 7 CONCLUSION

In this paper, we show our understanding of a current SOTA unsupervised skill discovery algorithms through the lens of MISL. Our analysis allowed the development of a simpler method CSF, which performs on par with METRA in most settings. More broadly, we provide evidence that mutual information maximization can still be effective to build high performing skill discovery algorithms.

**Limitations.** While CSF performs relatively well on the standard benchmarks, it is unclear how to *scale* the performance to increasingly complex environments such as Craftax (Matthews et al., 2024) or VIMA (Jiang et al., 2022), which present an increased number of objects, partial observability, stochasticity, and discrete action spaces. Another open question is how to perform scalable pre-training on large datasets, e.g., BridgeData V2 (Walke et al., 2023) or YouCook2 (Zhou et al., 2018), using MISL algorithms such as CSF to get both transferable state representations and diverse skill-conditioned policies. We leave investigating these empirical scaling limits to future work.

## REPRODUCIBILITY STATEMENT

We have made our code publicly available here: https://github.com/Princeton-RL/contrastive-successor-features. In addition, we have dedicated several sections in the appendix to further ensure reproducibility of our results. Appendix B provides a detailed account of all experimental details, including GPU types, training times, hyperparameters and architectural details (Tables 1, 2, and 3), and detailed descriptions of the parameters of the various settings we use to compare with prior work (Appendices B.3, B.4, B.5). Finally, we have also included full proofs for the theoretical results stated in Proposition 1, 2, and 3 which can be found in Appendices A.2, A.3, and A.4, respectively.

## ACKNOWLEDGEMENTS

We thank the National Science Foundation (Grant No. 2239363) for providing funding for this work. Any opinions, findings, conclusions, or recommendations expressed in this material are those of the author(s) and do not necessarily reflect the views of the National Science Foundation. We thank Seohong Park for providing code for several of the baselines in the paper. In addition, we thank Qinghua Liu for providing feedback on early drafts of this paper. We also thank Princeton Research Computing.

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

# A  THEORETICAL ANALYSIS

## A.1  MUTUAL INFORMATION MAXIMIZATION AS A MIN-MAX OPTIMIZATION PROBLEM

Maximizing the mutual information $I^\pi(S, S'; Z)$ (Eq. 1) is more challenging than standard RL because the reward function $\log p^\pi(z \mid s, s')$ depends on the policy itself. To break this cyclic dependency, we introduce a variational distribution $q(z \mid s, s') \in Q \triangleq \{q(z \mid s, s')\}$ to approximate the posterior $p^\pi(z \mid s, s')$, where we assume that the variational family $Q$ is expressive enough to cover the ground true distribution under any $\pi$:

**Assumption 1.** *For any skill-conditioned policy $\pi : \mathcal{S} \times \mathcal{Z} \mapsto \Delta(\mathcal{A})$, there exists $q^\star(z \mid s, s') \in Q$ such that $q^\star(z \mid s, s') = p^\pi(z \mid s, s')$.*

This assumption allows us to rewrite Eq. 1 as

$$\max_\pi \mathbb{E}_{p^\pi(s,s',z)}[\log p^\pi(z \mid s, s')] - \min_{q \in Q} \mathbb{E}_{p^\pi(s,s')} \left[ D_{\mathrm{KL}} \left( p^\pi(\cdot \mid s, s') \parallel q(\cdot \mid s, s') \right) \right],$$

where $D_{\mathrm{KL}} \left( p^\pi(\cdot \mid s, s') \parallel q(\cdot \mid s, s') \right)$ is the KL divergence between distributions $p^\pi$ and $q$ and it satisfies $D_{\mathrm{KL}}(p^\pi(\cdot \mid s, s') \parallel q(\cdot \mid s, s')) = 0 \iff p^\pi(z \mid s, s') = q(z \mid s, s')$. The new max-min optimization problem can be solved iteratively by first choosing variational distribution $q(z \mid s, s')$ to fit the ground truth $p^\pi(z \mid s, s')$ and then choosing policy $\pi$ to maximize discounted return defined by the intrinsic reward $q(z \mid s, s')$:

$$q_{k+1} \leftarrow \arg\max_{q \in Q} \mathbb{E}_{p^{\pi_k}(s,s',z)} \left[ \log q(z \mid s, s') \right],$$

$$\pi_{k+1} \leftarrow \arg\max_\pi \mathbb{E}_{p^\pi(s,s',z)}[\log q_k(z \mid s, s')],$$

where $k$ indicates the number of updates. In practice, the data used to update $q$ are uniformly sampled from a replay buffer typically containing trajectories from historical policies. Thus, the behavioral policy is exactly the average of historical policies $\beta = \frac{1}{k} \sum_{i=1}^k \pi_i(a \mid s, z)$ and the update rule for $q$ becomes

$$q_{k+1} \leftarrow \arg\max_{q \in Q} \mathbb{E}_{p^\beta(s,s',z)}[\log q(z \mid s, s')].$$

## A.2  PROOF OF PROPOSITION 1

**Proposition 1.** *The optimal state representation $\phi^\star$ of the actual METRA representation objective (Eq. 7) satisfies*

$$\mathbb{E}_{p^\beta(s,s')} \left[ \|\phi^\star(s') - \phi^\star(s)\|_2^2 \right] = 1.$$

*Proof.* Suppose that the optimal $\phi^\star$ satisfies

$$0 \leq \mathbb{E}_{p^\beta(s,s')} \left[ \|\phi^\star(s') - \phi^\star(s)\|_2^2 \right] = \alpha^2 < 1, \tag{12}$$

where $0 \leq \alpha < 1$. Then, there exists a $1/\alpha > 1$ that scales the expectation in Eq. 12 to exactly 1:

$$\frac{1}{\alpha^2} \mathbb{E}_{p^\beta(s,s')} \left[ \|\phi^\star(s') - \phi^\star(s)\|_2^2 \right] = 1.$$

Note that, when $\mathbb{E}_{p(z)p^\beta(s,s'|z)} \left[ (\phi^\star(s') - \phi^\star(s))^\top z \right] \geq 0$, the $\phi^\star/\alpha$ will also scale the objective to a larger number

$$\frac{1}{\alpha} \mathbb{E}_{p(z)p^\beta(s,s'|z)} \left[ (\phi^\star(s') - \phi^\star(s))^\top z \right] \geq \mathbb{E}_{p(z)p^\beta(s,s'|z)} \left[ (\phi^\star(s') - \phi^\star(s))^\top z \right],$$

which contradicts the assumption that $\phi^\star$ is optimal. When $\mathbb{E}_{p(z)p^\beta(s,s'|z)} \left[ (\phi^\star(s') - \phi^\star(s))^\top z \right] < 0$, taking $-\phi^\star/\alpha$ gives us the same result. Therefore, we conclude that the optimal $\phi^\star$ must satisfy $\mathbb{E}_{p^\beta(s,s')} \left[ \|\phi^\star(s') - \phi^\star(s)\|_2^2 \right] = 1$. □

## A.3 PROOF OF PROPOSITION 2

**Proposition 2.** *There exists a $\lambda_0(d)$ depending on the dimension $d$ of the state representation $\phi$ such that the following second-order Taylor approximation holds*

$$\lambda_0(d)(1 - \mathbb{E}_{p^\beta}\left[\|\phi(s') - \phi(s)\|_2^2\right]) \approx LB_-^\beta(\phi).$$

*Proof.* We first compute $\log \mathbb{E}_p(z)\left[e^{(\phi(s') - \phi(s))^\top z}\right]$ analytically,

$$
\begin{aligned}
\log \mathbb{E}_p(z)\left[e^{(\phi(s') - \phi(s))^\top z}\right] &= \log C_d(0) \int e^{(\phi(s') - \phi(s))^\top z} dz \\
&= \log \frac{C_d(0)}{C_d(\|\phi(s') - \phi(s)\|_2)} \\
&\quad + \log \int C_d(\|\phi(s') - \phi(s)\|_2) e^{\|\phi(s') - \phi(s)\|_2 \frac{(\phi(s') - \phi(s))^\top z}{\|\phi(s') - \phi(s)\|_2}} dz \\
&\overset{(a)}{=} \log \frac{C_d(0)}{C_d(\|\phi(s') - \phi(s)\|_2)} \\
&= \log \frac{\Gamma(d/2)(2\pi)^{d/2}\mathcal{I}_{d/2-1}(\|\phi(s') - \phi(s)\|_2)}{2\pi^{d/2}\|\phi(s') - \phi(s)\|_2^{d/2-1}} \\
&= \log \frac{\Gamma(d/2)2^{d/2-1}\mathcal{I}_{d/2-1}(\|\phi(s') - \phi(s)\|_2)}{\|\phi(s') - \phi(s)\|_2^{d/2-1}},
\end{aligned}
\tag{13}
$$

where $\Gamma(\cdot)$ is the Gamma function, $\mathcal{I}_v(\cdot)$ denotes the modified Bessel function of the first kind at order $v$, $C_d(\cdot)$ denotes the normalization constant for $d$-dimensional von Mises-Fisher distribution, and in *(a)* we use the definition of the density of vMF distributions. Applying Taylor expansion (Abramowitz & Stegun, 1968) to Eq. 13 around $\|\phi(s') - \phi(s)\|_2 = 0$ by using Mathematica (Inc., 2024) gives us a polynomial approximation

$$\log \frac{\Gamma(d/2)2^{d/2-1}\mathcal{I}_{d/2-1}(\|\phi(s') - \phi(s)\|_2)}{\|\phi(s') - \phi(s)\|_2^{d/2-1}} = \frac{1}{2d}\|\phi(s') - \phi(s)\|_2^2 + O(\|\phi(s') - \phi(s)\|_2^3)$$

Now we can simply set $\lambda_0(d) = \frac{1}{2d}$ to get

$$\lambda_0(d)(1 - \|\phi(s') - \phi(s)\|_2^2) \approx -\log \mathbb{E}_p(z)\left[e^{(\phi(s') - \phi(s))^\top z}\right] + \text{const.}.$$

Hence, we conclude that $\lambda_0(d)(1 - \mathbb{E}_{p^\beta(s,s')}\left[\|\phi(s') - \phi(s)\|_2^2\right])$ is a second-order Taylor approximation of $LB_-^\beta(\phi) = -\mathbb{E}_{p^\beta(s,s')}\left[\log \mathbb{E}_p(z)\left[e^{(\phi(s') - \phi(s))^\top z}\right]\right]$ around $\|\phi(s') - \phi(s)\|_2^2 = 0$ up to a constant factor of $\lambda_0(d)$. $\square$

## A.4 PROOF OF PROPOSITION 3

**Proposition 3.** *The METRA actor objective is a lower bound on the information bottleneck $I^\pi(S, S'; Z) - I^\pi(S, S'; \phi(S') - \phi(S))$, i.e., $J(\pi) \leq I^\pi(S, S'; Z) - I^\pi(S, S'; \phi(S') - \phi(S))$.*

*Proof.* We consider the mutual information between transition pairs and skills under the target policy $I^\pi(S, S'; Z)$. The standard variational lower bound (Barber & Agakov, 2004; Poole et al., 2019) of $I^\pi(S, S'; Z)$ can we written as:

$$I^\pi(S, S'; Z) \geq h(Z) + \mathbb{E}_{p^\pi(s,s',z)}[\log \tilde{q}(z \mid s, s')],$$

where $\tilde{q}(z \mid s, s')$ is an arbitrary variational approximation of $p^\pi(z \mid s, s')$. We can set $\log \tilde{q}(z \mid s, s')$ to be

$$\log \tilde{q}(z \mid s, s') = f(s, s', z) + \log p(z) - \log \mathbb{E}_{p(z)}\left[e^{f(s, s', z)}\right],$$

resulting in a lower bound:

$$I^\pi(S, S'; Z) \geq \underbrace{\mathbb{E}_{p^\pi(s,s',z)}[(\phi(s') - \phi(s))^\top z]}_{\text{LB}^\pi_+(\phi)} - \underbrace{\mathbb{E}_{p^\pi(s,s')}\left[\log \mathbb{E}_{p(z)}\left[e^{(\phi(s') - \phi(s))^\top z}\right]\right]}_{\text{LB}^\pi_-(\phi)},$$

where $\text{LB}^\pi_+(\phi)$ is exactly the same as the RL objective $J(\pi)$ (Eq. 6). This lower bound is similar to Eq. 8 but it is under the target policy $\pi$ instead.

Equivalently, we can write the RL objective as

$$J(\pi) = \mathbb{E}_{p(z)p^\pi(s_+=s|z)p^\pi(s'|s,z)}\left[(\phi(s') - \phi(s))^\top z {\color{orange}- \log \mathbb{E}_{p(z')}\left[e^{(\phi(s') - \phi(s))^\top z'}\right]}\right.$$
$$\left.{\color{orange}+ \log \mathbb{E}_{p(z')}\left[e^{(\phi(s') - \phi(s))^\top z'}\right]}\right] = \text{LB}^\pi(\phi) - \text{LB}^\pi_-(\phi)$$

where the two log-expected-exps cancel with each other. We next focus on the additional $\text{LB}^\pi_-(\phi) = -\mathbb{E}_{p(z)p^\pi(s_+=s|z)p^\pi(s'|s,z)}\left[\log \mathbb{E}_{p(z')}\left[e^{(\phi(s') - \phi(s))^\top z'}\right]\right]$, which can be interpreted as a resubstitution entropy estimator of $\phi(s') - \phi(s)$ (Wang & Isola, 2020; Ahmad & Lin, 1976):

$$\text{LB}^\pi_-(\phi) = -\mathbb{E}_{p^\pi(s,s')}\left[\log \mathbb{E}_{p(z)}\left[e^{(\phi(s') - \phi(s))^\top z}\right]\right]$$

$$\overset{(a)}{=} -\frac{1}{N}\sum_{i=1}^N\left[\log\left(\frac{1}{N}\sum_{j=1}^N C_d(\|\phi(s_i') - \phi(s_i)\|_2)e^{\|\phi(s_i') - \phi(s_i)\|_2 \frac{(\phi(s_i') - \phi(s_i))^\top z_j}{\|\phi(s_i') - \phi(s_i)\|_2}}\right)\right.$$
$$\left. + \log\frac{1}{C_d(\|\phi(s_i') - \phi(s_i)\|_2)}\right]$$

$$= -\frac{1}{N}\sum_{i=1}^N\left(\log\hat{p}_{\text{vMF-KDE}}(\phi(s_i') - \phi(s_i)) + \log\frac{1}{C_d(\|\phi(s_i') - \phi(s_i)\|_2)}\right)$$

$$= \hat{h}^\pi(\phi(S') - \phi(S)) - \mathbb{E}_{p^\pi(s,s')}\left[\log\frac{1}{C_d(\|\phi(s_i') - \phi(s_i)\|_2)}\right]$$

$$\overset{(b)}{=} \hat{I}^\pi(S, S'; \phi(S') - \phi(S)) - \mathbb{E}_{p^\pi(s,s')}\left[\log\frac{1}{C_d(\|\phi(s_i') - \phi(s_i)\|_2)}\right]$$

$$\overset{(c)}{\approx} \hat{I}^\pi(S, S'; \phi(S') - \phi(S)) - \lambda_0(d)\mathbb{E}_{p^\pi(s,s')}\left[\|\phi(s') - \phi(s)\|_2^2\right] + \text{const.}$$

$$\overset{(d)}{\approx} \hat{I}^\pi(S, S'; \phi(S') - \phi(S)) + \text{const.},$$

where $C_d(\cdot)$ denotes the normalization constant for $d$-dimensional von Mises-Fisher distribution, in $(a)$ we use Monte Carlo estimator with $N$ transitions and skills $\{(s_i, s_i', z_i)\}_{i=1}^N$ to rewrite the expectation, in $(b)$ we replace the entropy estimator $\hat{h}^\pi$ with the mutual information estimator $\hat{I}^\pi$ since $\phi(s') - \phi(s)$ is a deterministic function of $(s, s')$, in $(c)$ we apply the same approximation in Prop. 2, and in $(d)$ the expected squared norm is replaced by $1.0$, assuming that Prop. 1 holds. Taken together, we conclude that maximizing the RL objective $J(\pi)$ is approximately equivalent to maximizing a lower bound on the information bottleneck $I^\pi(S, S'; Z) - I^\pi(S, S'; \phi(S') - \phi(S))$. □

### A.5 Mutual Information Objectives used in Prior Methods

Prior MISL methods (Eysenbach et al., 2019; Gregor et al., 2016; Sharma et al., 2020; Hansen et al., 2020; Campos et al., 2020) adopt the min-max optimization procedure (Eq. 2 & 3) and use the same functional form of the lower bound on different variants of the mutual information as their objectives. We elaborate which mutual information each prior method optimizes next.

For DIAYN (Eysenbach et al., 2019) and VISR (Hansen et al., 2020), both representation learning and policy learning objectives are (up to some constants) lower bounds on $I(S; Z) \gtrsim -\mathbb{E}_{p(z)}[\log p(z)] + \mathbb{E}_{p(s,z)}[\log q(z \mid s)]$, where $\gtrsim$ denotes $>$ up to constant scaling or shifting. See Eq. 2 & 3 of (Eysenbach et al., 2019) and Eq. 9 of (Hansen et al., 2020) for details.

For VIC (Gregor et al., 2016), both representation learning and policy learning objectives are lower bounds on $I(S_T; Z \mid S_0) \geq -E_{p(z|s_0)}[\log p(z \mid s_0)] + \mathbb{E}_{p(s_T,z|s_0)}[\log q(z \mid s_0, s_T)]$, where $S_0$

---

**Algorithm 2** A General Mutual Information Skill Learning Framework

---

1: **Input** state representations $\phi : \mathcal{S} \mapsto \mathbb{R}^d$, latent skill distribution $p(z)$, and skill-conditioned policy $\pi : \mathcal{S} \times \mathcal{Z} \mapsto \Delta(\mathcal{A})$.
2: **for** each iteration **do**
3:     Collect trajectory $\tau$ with $z \sim p(z)$ and $a \sim \pi(a \mid s, z)$, and then add $\tau$ to the replay buffer.
4:     Sample $B = \{(s, s', z)\}$ from the replay buffer.
5:     Update $\phi$ by maximizing a lower bound on $I^\beta(S, S'; Z)$ constructed using $B$.
6:     Relabel the intrinsic reward as a lower bound on $I^\pi(S, S'; Z) - I^\pi(S, S'; \phi(S') - \phi(S))$.
7:     Update $\pi$ using an off-policy RL algorithm with $B \cup \{r(s, s', z)\}$.
8: **Return** $\phi^\star$ and $\pi^\star$.

---

denotes the random variable for initial states and $S_T$ denotes the random variable for terminal states. See Eq. 3 of (Gregor et al., 2016) for details.

For DADS (Sharma et al., 2020), both representation learning and policy learning objectives are (approximate) lower bounds on $I(S'; Z \mid S) \geq -\mathbb{E}_{p(s'|s)}[\log p(s' \mid s)] + \mathbb{E}_{p(s',z|s)}[\log q(s' \mid s, z)]$. See Eq. 5 & 6 of (Sharma et al., 2020) for details.

### A.6 THE EFFECT OF THE SCALING COEFFICIENT $\xi$

In Sec. 5.1, we introduce a coefficient $\xi$ to scale the negative term $\mathrm{LB}^\beta_-(\phi)$ in the contrastive lower bound and find that a proper choice of $\xi$ improves the performance of CSF empirically (Appendix C.3). This coefficient has an effect similar to the tradeoff coefficient used in information bottleneck optimization ($\beta$ in (Alemi et al., 2017)). We also note that when setting $\xi \geq 1$, the new representation objective $\mathrm{LB}^\beta_+(\phi) + \xi \mathrm{LB}^\beta_-(\phi)$ is still a (scaled) contrastive lower bound on the mutual information $I^\beta(S, S'; Z)$. The reason is that $\mathrm{LB}^\beta_-(\phi)$ is always non-positive:

$$\begin{aligned}
\mathrm{LB}^\beta_-(\phi) &= -\mathbb{E}_{p^\beta(s,s')}\left[\log \mathbb{E}_{p(z')}\left[e^{(\phi(s')-\phi(s))^\top z'}\right]\right] \\
&\overset{(a)}{\leq} -\mathbb{E}_{p^\beta(s,s')}\left[\log e^{(\phi(s')-\phi(s))^\top \mathbb{E}_{p(z')}[z']}\right] \\
&\overset{(b)}{=} -\mathbb{E}_{p^\beta(s,s')}\left[\log e^0\right] \\
&= 0,
\end{aligned}$$

where in *(a)* we apply Jensen's inequality and in *(b)* we use the symmetry of $\mathrm{UNIF}(\mathbb{S}^{d-1})$. Therefore, for any $\xi \geq 1$, we have $\mathrm{LB}^\beta_+(\phi) + \xi \mathrm{LB}^\beta_-(\phi) \leq \mathrm{LB}^\beta_+(\phi) + \mathrm{LB}^\beta_-(\phi)$.

### A.7 A GENERAL MUTUAL INFORMATION SKILL LEARNING FRAMEWORK

The general mutual information skill learning algorithm alternates between *(1)* collecting data, *(2)* learning state representation $\phi$ by maximizing a lower bound on the mutual information $I^\beta(S, S'; Z)$ under the behavioral policy $\beta$, *(3)* relabeling the intrinsic reward as a lower bound on the information bottleneck $I^\pi(S, S'; Z) - I^\pi(S, S'; \phi(S') - \phi(S))$, and finally *(4)* using an off-the-shelf off policy RL algorithm to learning the skill-conditioned policy $\pi$. We show the pseudo-code of this algorithm in Alg. 2.

### A.8 CONNECTION BETWEEN REPRESENTATIONS LEARNED BY METRA AND CONTRASTIVE REPRESENTATIONS

In our experiments (Sec. 6.2), we sample 10K pairs of $(s, s', z)$ from the replay buffer and use them to visualize the histograms of conditional differences in representations $\phi(s') - \phi(s) - z$ and normalized marginal differences in representations $(\phi(s') - \phi(s))/\|\phi(s') - \phi(s)\|_2$. The resulting histograms (Fig. 2 *(Center)* & *(Right)*) indicate two intriguing properties of representations learned by METRA. First, given a set of skills $\{z\}$, the differences in representations subtracting the corresponding skills $\phi(s') - \phi(s) - z$ converges to an isotropic Gaussian distribution:

**Claim 1.** *The state representations $\phi$ learned by METRA satisfies that $\phi(s') - \phi(s) - z \xrightarrow{d} \mathcal{N}(0, \sigma_\phi^2 I)$, or, equivalently, $\phi(s') - \phi(s) \mid z \xrightarrow{d} \mathcal{N}(z, \sigma_\phi^2 I)$, where $\xrightarrow{d}$ denotes convergence in distribution and $\sigma_\phi$ is the standard deviation of the isotropic Gaussian.*

Second, taking the marginal over all possible skills, the normalized difference in representations $(\phi(s') - \phi(s))/\|\phi(s') - \phi(s)\|_2$ converges to a uniform distribution on the $d$-dimensional unit hypersphere $\mathbb{S}^{d-1}$:

**Claim 2.** *The state representations $\phi$ learned by METRA also satisfy $\frac{\phi(s') - \phi(s)}{\|\phi(s') - \phi(s)\|_2} \xrightarrow{d} \mathrm{UNIF}(\mathbb{S}^{d-1})$.*

We next propose a Lemma that relates a isotropic Gaussian distribution to a von Mises–Fisher distribution (Wikipedia, 2024) and then draw the connection between Claim 1 and Claim 2.

**Lemma 1.** *Given an $n$-dimensional isotropic Gaussian distribution $\mathcal{N}(\mu, \sigma^2 I)$ with $\|\mu\|_2 = r_\mu$, a von Mises–Fisher distribution $\mathrm{vMF}\left(\mu/r_\mu, r_\mu/\sigma^2\right)$ can be obtained by restricting the support to be a hypersphere with radius $r_\mu$, i.e., $\{x : \|x\|_2 = r_\mu\}$.*

*Proof.* The probability density function of $\mathcal{N}(\mu, \sigma^2 I)$ be written as

$$
\begin{aligned}
p(x) &= \frac{1}{(2\pi\sigma)^{\frac{n}{2}}} \exp\left(-\frac{1}{2\sigma^2}(x-\mu)^\top(x-\mu)\right) \\
&= \frac{1}{(2\pi\sigma)^{\frac{n}{2}}} \exp\left(-\frac{1}{2\sigma^2}\left(\|x\|_2^2 - 2x^\top\mu + \|\mu\|_2^2\right)\right)
\end{aligned}
$$

When conditioning on $\|x\|_2 = r_\mu$, we have

$$
\begin{aligned}
\frac{1}{(2\pi\sigma)^{\frac{n}{2}}} \exp\left(-\frac{1}{2\sigma^2}\left(\|x\|_2^2 - 2x^\top\mu + \|\mu\|_2^2\right)\right) &= \frac{1}{(2\pi\sigma)^{\frac{n}{2}}} \exp\left(-\frac{1}{2\sigma^2}\left(2r_\mu^2 - 2x^\top\mu\right)\right) \\
&= \frac{1}{(2\pi\sigma)^{\frac{n}{2}}} \exp\left(\frac{r_\mu}{\sigma^2} \cdot \frac{\mu^\top x}{r_\mu} - \frac{r_\mu^2}{\sigma^2}\right) \\
&\propto \exp\left(\frac{r_\mu}{\sigma^2} \cdot \frac{\mu^\top x}{r_\mu}\right).
\end{aligned}
$$

After recomputing the normalizing constant, we recover the probability density function of the von Mises-Fisher distribution $\mathrm{vMF}\left(\mu/r_\mu, r_\mu/\sigma^2\right)$. $\qquad\square$

Since the didactic experiments in Sec. 6.2 have shown that $\phi(s') - \phi(s) \mid z$ converges to a Gaussian distribution $\mathcal{N}(z, \sigma_\phi^2 I)$ (Claim 1) and note that $\|z\|_2 = 1$, by applying Lemma A.8, we conjecture that restricting $\phi(s') - \phi(s)$ within $\{\|\phi(s') - \phi(s)\|_2 = 1\}$ produces a von Mises-Fisher distribution, i.e., $\frac{\phi(s') - \phi(s)}{\|\phi(s') - \phi(s)\|_2} \mid z \xrightarrow{d} \mathrm{vMF}(z, 1/\sigma_\phi^2)$. Furthermore, we can derive the marginal density of $\frac{\phi(s') - \phi(s)}{\|\phi(s') - \phi(s)\|_2}$,

$$
\begin{aligned}
p\left(\frac{\phi(s') - \phi(s)}{\|\phi(s') - \phi(s)\|_2}\right) &= \int p(z) p\left(\frac{\phi(s') - \phi(s)}{\|\phi(s') - \phi(s)\|_2} \,\Big|\, z\right) dz \\
&\overset{(a)}{=} C_d(0) \int C_d\left(\frac{1}{\sigma_\phi^2}\right) \exp\left(\frac{1}{\sigma_\phi^2} \cdot \frac{(\phi(s') - \phi(s))^\top z}{\|\phi(s') - \phi(s)\|_2}\right) dz \\
&= C_d(0),
\end{aligned}
$$

where in *(a)* we use the symmetric property of the density function of von Mises-Fisher distributions. Crucially, the marginal density indicates that $\frac{\phi(s') - \phi(s)}{\|\phi(s') - \phi(s)\|_2}$ follows a uniform distribution $\mathrm{UNIF}(\mathbb{S}^{d-1})$, which is exactly the observation in our experiments (Claim 2).

### A.9 INSIGHTS FOR THE INNER PRODUCTION CRITIC PARAMETERIZATION

Our ablation studies in Sec. 6.3 shows that the inner product parameterization of the critic $f(s, s', z) = (\phi(s') - \phi(s))^\top$ is important for CSF (Fig. 3 *(Right)*). There are two explanations for these observations. First, the inner product parameterization tries to push together the difference of the representation of transitions $\phi(s') - \phi(s)$ and the corresponding skill $z$, instead of focusing on representations of individual states $\phi(s)$ or $\phi(s')$. This intuition is inline with the observation from prior work that mutual information $I(S; Z)$ is invariant to bijective mappings on $S$ (See Fig. 2 of (Park et al., 2024)), e.g., translation and scaling, indicating that maximizing the mutual information between change of states and skills ($I(S, S'; Z)$) encourages better state space coverage. Second, the inner product parameterization allows us to analytically compute the second-order Taylor approximation in Proposition 2, drawing the connection between the METRA representation objective and contrastive learning.

### A.10 INTUITION FOR ZERO-SHOT GOAL INFERENCE

In zero-shot goal reaching setting, we want to figure out the corresponding latent skill $z$ when given a goal state or image $g$; a problem which can be cast as inferring the $z \in \mathcal{Z}$ that maximizes the posterior $p^\pi(z \mid s, g)$. Since the ground truth posterior $p^\pi(z \mid s, g)$ is unknown, a typical workaround is first estimating a variational approximation of $p^\pi(z \mid s, g)$ and then maximizing the variational posterior $q(z \mid s, g)$. We provide an intuition for zero-shot goal inference by specifying the variational posterior as

$$q(z \mid s, g) \triangleq \frac{p(z)e^{(\phi(g)-\phi(s))^\top z}}{\mathbb{E}_{p(z)}\left[e^{(\phi(g)-\phi(s))^\top z'}\right]}$$

and solving the optimization problem in the latent skill space $\mathcal{Z}$

$$\arg\max_{z \in \mathcal{Z}} \ \log q(z \mid s, g),$$

or equivalently,

$$\arg\max_{z} \ (\phi(g) - \phi(s))^\top z \quad \text{s.t. } \|z\|_2^2 = 1.$$

Taking derivative of the Lagrangian and setting it to zero, the analytical solution is exactly $z^\star = \frac{\phi(g)-\phi(s)}{\|\phi(g)-\phi(s)\|_2}$, suggesting that the heuristic used by prior methods and our algorithm can be understood as a maximum a posteriori (MAP) estimation.

## B EXPERIMENTAL DETAILS

All experiments were run on a combination of GPUs consisting of NVIDIA GeForce RTX 2080 Ti, NVIDIA RTX A5000, NVIDIA RTX A6000, and NVIDIA A100. All experiments took at most 1 day to run to completion.

### B.1 SIMPLICITY OF CSF COMPARED TO METRA

The main differences between our method and METRA are (1) directly using the contrastive lower bound on the mutual information $I^\beta(S, S'; Z)$ as the representation objective, and (2) learning a policy by estimating the successor features, which is a vector-valued critic, instead of a scalar Q in an actor-critic style. Our method can be implemented based on METRA by making three changes (see algorithm summary in Sec. 5.2). Since the policy learning step only requires changing the output dimension of neural networks, CSF reduces the number of hyperparameters in the representation learning step compared to METRA. Next, we provide a hyperparameter comparison between CSF and METRA.

On the one hand, since our algorithm prevents the dual gradient descent optimization in the METRA representation objective, we do not have the slack variable $\epsilon$, do not have to specify the initial value of the dual variable $\lambda$, and remove the dual variable optimizer with its learning rate, resulting in three less hyperparameters. On the other hand, we introduce one coefficient $\xi$ to scale the negative term

Table 1: **CSF hyperparameters for unsupervised pretraining.**

| Hyperparameter | Value |
|---|---|
| Learning rate | 0.0001 |
| Horizon | 200, except for 50 in `Kitchen` |
| Parallel workers | 8, except for 10 in `Robobin` |
| State normalizer | used in state-based environments only |
| Replay buffer batch size | 256 |
| Gradient updates per trajectory collection round | 50 (`Ant`, `Cheetah`), 200 (`Humanoid`, `Quadruped`), 100 (`Kitchen`, `Robobin`) |
| Frame stack | 3 for image-based, n/a for state-based |
| Trajectories per data collection round | 8, except for 10 in `Robobin` |
| Automatic entropy tuning | yes |
| $\xi$ (scales second term in Eq. 10) | 5 |
| Number of negative $z$s to compute $\text{LB}_-(\phi)$ | 256 (in-batch negatives) |
| EMA $\tau$ (target network) | $5e^{-3}$ |
| $\phi, \pi, \psi$ network hidden dimension | 1024 |
| $\phi, \pi, \psi$ network number of layers | 1 input, 1 hidden, 1 output |
| $\phi, \pi, \psi$ network nonlinearity | relu ($\phi$), tanh ($\pi$), relu ($\psi$) |

$\text{LB}_-^{\beta}(\phi)$ in the contrastive lower bound, which has an effect similar to the tradeoff coefficient used in information bottleneck optimization ($\beta$ in (Alemi et al., 2017)). Taken together, CSF uses three less hyperparameters than METRA and introduces one hyperparameter to balance the positive term ($\text{LB}_+^{\beta}(\phi)$) and the negative term ($\text{LB}_-^{\beta}(\phi)$) in the representation objective.

## B.2 EXPERIMENTAL SETUP

**Environments.** We choose to evaluate on the following six tasks: `Ant` and `HalfCheetah` from Gym (Todorov et al., 2012; Brockman et al., 2016), `Quadruped` and `Humanoid` from DeepMind Control (DMC) Suite (Tassa et al., 2018), and `Kitchen` and `Robobin` from LEXA (Mendonca et al., 2021). We choose these six tasks to be consistent with the original METRA work (Park et al., 2024). In addition, we added `Robobin` as another manipulation task since the original five tasks are all navigation tasks except for `Kitchen`. The observations are state-based in `Ant` and `HalfCheetah` and $64 \times 64$ RGB images of the scene in all other tasks.

**Baselines.** We consider five baselines. **(1) METRA (Park et al., 2024)** is the state-of-the-art approach which provides the motivation for deriving CSF. **(2) CIC (Laskin et al., 2022)** uses a rank-based contrastive loss (InfoNCE) to learn representations of transitions and then maximizes a state entropy estimate constructed using these representations. **(3) DIAYN (Eysenbach et al., 2019)** represents a broad category of methods that first learn a parametric discriminator $q(z \mid s, s')$ (or $q(z \mid s)$) to predict latent skills from transitions and then construct the reverse variational lower bound on mutual information (Campos et al., 2020) as an intrinsic reward. **(4) DADS (Sharma et al., 2020)** builds upon the forward variational lower bound on mutual information (Campos et al., 2020) which requires maximizing the state entropy $h(S)$ to encourage state coverage while minimizing the conditional state entropy $h(S \mid Z)$ to distinguish different skills. There is a family of methods studying variational approximations of $h(S)$ and $h(S \mid Z)$ (Campos et al., 2020; Liu & Abbeel, 2021; Laskin et al., 2022; Lee et al., 2019; Sharma et al., 2020) of which DADS is a representative. **(5) VISR (Hansen et al., 2020)** is similar to DIAYN in that it also trains the representations $\phi$ by learning a discriminator to maximize the likelihood of a skill given a state, though the discriminator is parametrized as a vMF distribution. In addition, VISR learns successor features that allow it to perform GPI as well as fast task adaptation after unsupervised pretraining. Note that our version of VISR does not include GPI since we evaluate on continuous control environments.

Table 2: **Skill dimensions per method and environment.** We list the skill dimension for all methods and environments reported in the paper.

|      | Ant | HalfCheetah | Quadruped | Humanoid | Kitchen | Robobin |
|------|-----|-------------|-----------|----------|---------|---------|
| **CSF**   | 2  | 2  | 4  | 8  | 4  | 9  |
| **METRA** | 2  | 16 | 4  | 2  | 24 | 9  |
| **DIAYN** | 50 | 50 | 50 | 50 | 50 | 50 |
| **DADS**  | 3  | 3  | -  | -  | -  | -  |
| **CIC**   | 64 | 64 | 64 | 64 | 64 | 64 |
| **VISR**  | 5  | 5  | 5  | 5  | 5  | 5  |

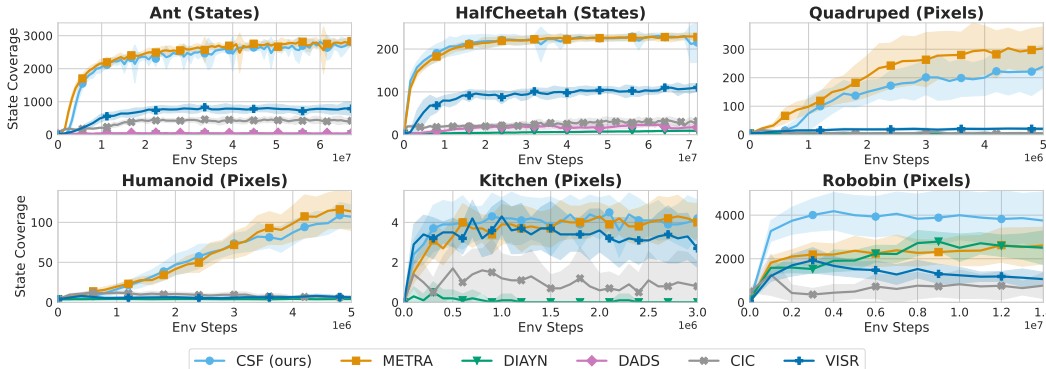

Figure 5: **State space coverage.** We plot the unique number of coordinates visited by the agent, except for Kitchen where we plot the task coverage. We find CSF matches the prior state-of-the-art MISL algorithms on $4/6$ tasks, and strongly outperforms METRA in Robobin. Shaded areas indicate one standard deviation.

## B.3 EXPLORATION PERFORMANCE

Please see Fig. 5 for the full set of exploration results. We can see that CSF continues to perform on par with METRA, while sometimes outperforming METRA (Robobin) and sometimes underperforming METRA (Quadruped).

For CSF, all tasks were trained with continuous $z$ sampled from a uniform vMF distribution and $\lambda = 5$. METRA also uses a continuous $z$ sampled from a uniform vMF distribution for all environments except for HalfCheetah and Kitchen, where we used a one-hot discrete $z$, consistent with the original work (Park et al., 2024). CIC uses a continuous $z$ sampled from a standard Gaussian for all environments. DIAYN uses a one-hot discrete $z$ for all environments. DADS uses a continuous $z$ sampled from a uniform distribution on $[-1, 1]$ for all environments. Finally, VISR uses a continuous $z$ sampled from a uniform vMF distribution for all environments. Please refer to Table 2 for a full overview of skill dimensions per method and environment. A table with all relevant hyperparameters for the unsupervised training phase can be found in Table 1.

## B.4 ZERO-SHOT GOAL REACHING

Please see Fig. 6 for the full set of goal reaching results. We find CSF to generally perform closely to METRA, though slightly underperforming in Quadruped, Humanoid, and Kitchen. In Ant however, CSF outperforms METRA.

**Goal sampling.** We closely follow the setup in Park et al. (2024). For all baselines, 50 goals are randomly sampled from $[-50, 50]$ in Ant, $[100, 100]$ in HalfCheetah, $[-15, 15]$ in Quadruped,

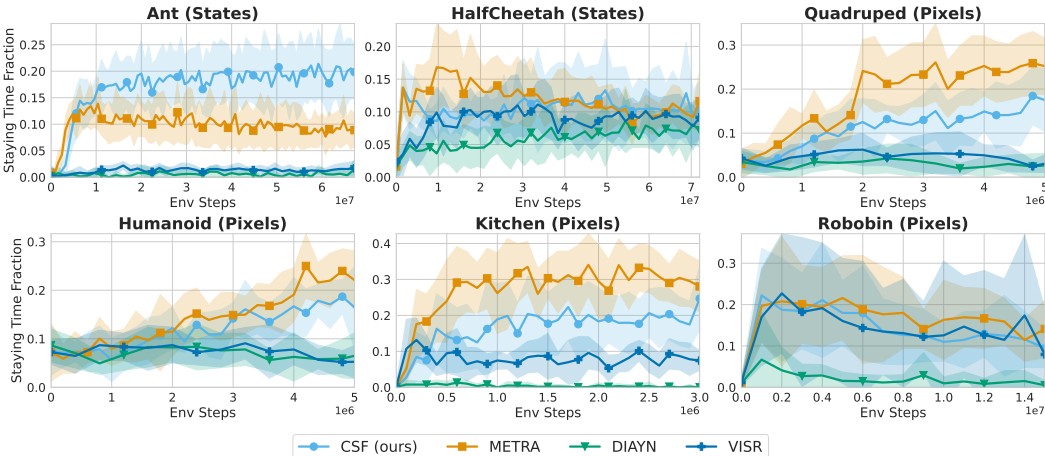

Figure 6: **Goal reaching.** We compare CSF with baselines on goal-reaching tasks. We find that CSF achieves strong performance on `Ant` and mostly outperforms DIAYN and VISR. However, CSF lags a bit behind METRA on `Quadruped`, `Kitchen`, and `Humanoid`. All means and standard deviations are computed across ten random seeds. Shaded areas indicate one standard deviation.

and $[-10, 10]$ in `Humanoid`. In `Kitchen`, we sample 50 times at random from the following built-in tasks: BottomBurner, LightSwitch, SlideCabinet, HingeCabinet, Microwave, and Kettle. In `Robobin`, we sample 50 times at random from the following built-in tasks: ReachLeft, ReachRight, PushFront, and PushBack.

**Evaluation**   Unlike prior methods (Park et al., 2022; 2024; Sharma et al., 2020; Mendonca et al., 2021), we choose the *staying time fraction* instead of the *success rate* as our evaluation metric. The staying time indicates the number of time steps that the agent stays at the goal divided by the horizon length, while the success rate simply indicates whether the agent reaches the goal at *any* time step. Importantly, a high success rate does not necessarily imply a high staying time fraction (e.g., the agent might overshoot the goal after success).

**Skill inference.**   Prior work (Park et al., 2022; 2024; 2023) has proposed a simple inference method by setting the skill to the difference in representations $z = \frac{\phi(g) - \phi(s)}{\|\phi(g) - \phi(s)\|_2}$, where $g$ indicates the goal. We choose to use the same approach for CSF and METRA and provide some theoretical intuition for this strategy in Appendix A.10. For DIAYN, we follow prior work (Park et al., 2024) and set $z = \text{one\_hot}[\arg\max_i q(z|g)_i]$.

### B.5   HIERARCHICAL CONTROL

Please see Fig. 7 for the full set of hierarchical control results. We find CSF to perform closely to METRA in most environments, though it outperforms METRA on `AntMultiGoal` and underperforms METRA on `QuadrupedGoal`. CSF outperforms all other baselines on all environments.

We use SAC (Haarnoja et al., 2018) for `AntMultiGoal`, `HumanoidGoal`, and `QuadrupedGoal`. We use PPO (Schulman et al., 2017) for `CheetahGoal` and `CheetahHurdle`. For all state-based environments, we initialize (and freeze) the child policy with a checkpoint trained with 64M environment steps. For image-based environments, we use checkpoints trained with 4.8M environments. A table with all relevant hyperparameters for training the hierarchical control policy can be found in Table 3.

Table 3: **CSF hyperparameters for hierarchical control.**

| Hyperparameter | Value |
|---|---|
| Learning rate | 0.0001 |
| Option timesteps length | 25 |
| Total horizon length | 200 |
| Parallel workers | 8 |
| Trajectories per data collection round | 8, except for `Cheetah` where we use 64 |
| Algorithm | SAC, except for `Cheetah` where we use PPO |
| State normalizer | used in state-based environments only |
| Replay buffer batch size | 256 |
| Gradient updates per trajectory collection round | 50, except for `Cheetah` where we use 10 |
| Frame stack | 3 for image-based, n/a for state-based |
| $\pi$ (parent, child) networks hidden dimension | 1024 |
| $\pi$ (parent, child) networks number of layers | 1 input, 1 hidden, 1 output |
| $\pi$ (parent, child) networks nonlinearity | tanh |
| Child policy frozen? | yes |

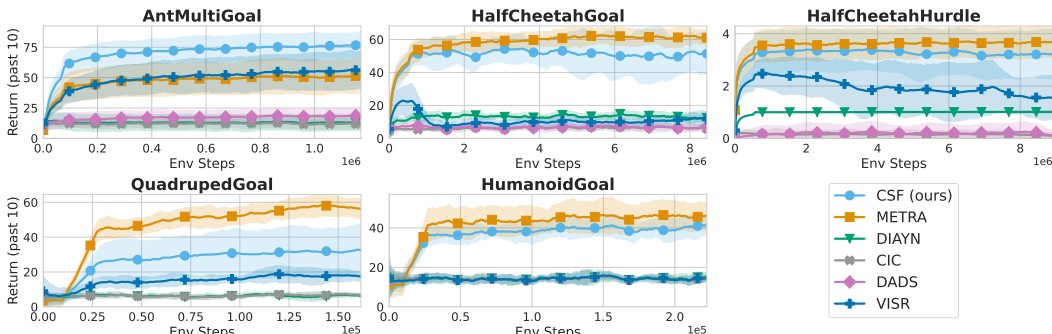

Figure 7: **Hierarchical control.** We compare CSF with baselines on hierarchical control tasks using returns averaged over the 10 past evaluations. We find CSF to perform mostly competitively compared to METRA, outperforming METRA in `AntMultiGoal`, but underperforming in `QuadrupedGoal` (and to a small extent in `HalfCheetahGoal` and `HumanoidGoal`). All means and standard deviations are computed across ten random seeds. Shaded areas indicate one standard deviation.

## C  ADDITIONAL EXPERIMENTS

### C.1  QUADRATIC APPROXIMATION OF $\mathrm{LB}_2^\beta(\phi)$

We conduct experiments to study the accuracy of the quadratic approximation in Prop. 2 in practice. To answer this question, we reuse the METRA algorithm trained on the didactic `Ant` environment and compare $\log \mathbb{E}_{p(z)}[e^{(\phi(s')-\phi(s))^\top z}]$ against $\|\phi(s') - \phi(s)\|_2^2$. We can compute $\log \mathbb{E}_{p(z)}[e^{(\phi(s')-\phi(s))^\top z}]$ analytically because $d = 2$ in our experiments. Results in Fig. 8 shows a clear linear relationship between $\log \mathbb{E}_{p(z)}[e^{(\phi(s')-\phi(s))^\top z}]$ and $\|\phi(s') - \phi(s)\|_2^2$, suggesting that the slope of the least squares linear regression is near the theoretical prediction, i.e., $\lambda_0(d) = \frac{1}{2d} = 0.25 \approx 0.2309$. We conjecture that this linear relationship still exists for higher dimensional $d$ and, therefore, the second-order Taylor approximation proposed by Prop. 2 is practical.

### C.2  METRA AND CSF ARE SENSITIVE TO THE SKILL DIMENSION

METRA leverages different skill dimensions for different environments. This caused us to investigate what the impact of the skill dimension on exploration performance is. In Fig. 9, we find that both

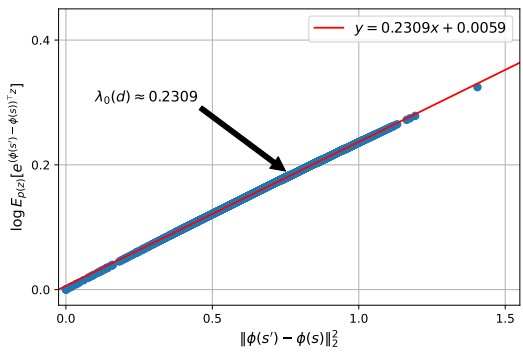
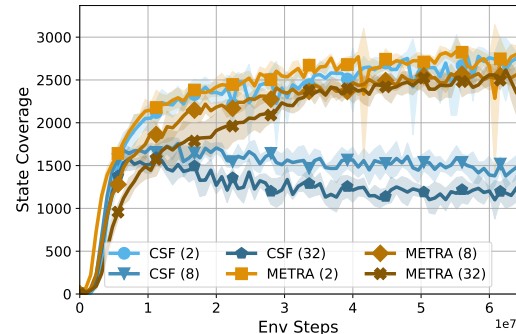

Figure 8: $\lambda_0(d)\|\phi(s') - \phi(s)\|_2^2$ is a second order Taylor approximation of $\log \mathbb{E}_{p(z)}[e^{(\phi(s')-\phi(s))^\top z}]$, where $\lambda_0(d) = \frac{1}{2d}$.

Figure 9: **Ablation of the skill dimension.** CSF and METRA (to a lesser extent) are sensitive to the dimension of skill (indicated in parentheses).

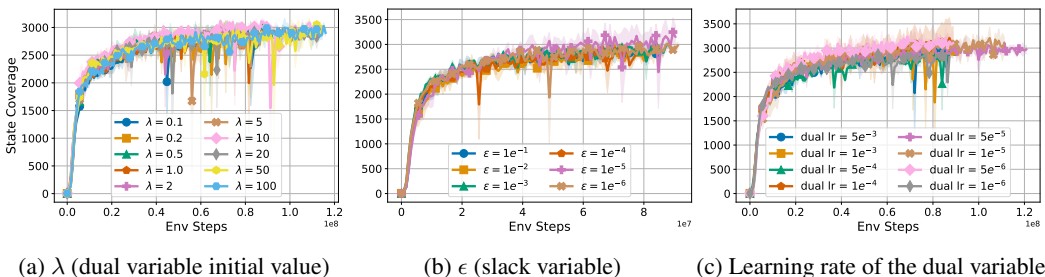

(a) $\lambda$ (dual variable initial value)     (b) $\epsilon$ (slack variable)     (c) Learning rate of the dual variable

Figure 11: METRA is robust to the initial value of the dual variable $\lambda$, the slack variable $\epsilon$, and the learning rate of the dual variable.

METRA (to a lesser extent) and CSF are quite sensitive to the skill dimension. We conclude that skill dimension is a key parameter to tune for practitioners when training their MISL algorithm.

### C.3 SENSITIVITY OF CSF TO THE SCALING COEFFICIENT $\xi$

We conduct ablation experiments to study the effect of the scaling coefficient $\xi$ on the negative term of the contrastive lower bound ($\mathrm{LB}_-^\beta(\phi)$ as well as investigating our theoretical prediction for selecting $\xi \geq 1$ in Appendix A.6. We compare the state coverage of different variants of CSF with $\xi$ choosing from $\{0.5, 1, 2, 5, 10\}$, plotting the mean and standard deviation over 5 random seeds. Results in Fig. 10 suggest that increasing $\xi$ to a higher values (2, 5, 10) boosts the state coverage of CSF, while a $\xi \leq 1$ hurts the performance. We choose to use $\xi = 5$ in our benchmark experiments.

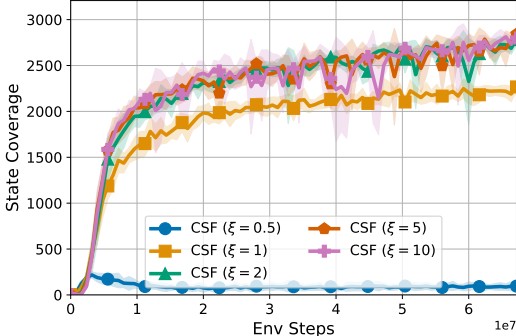

Figure 10: **The effect of $\xi$ in CSF.** Increasing $\xi$ to slightly higher values (2, 5, 10) boosts performance of CSF, while a $\xi < 1$ hurts performance substantially.

### C.4 SENSITIVITY OF METRA TO $\lambda$, $\epsilon$, AND THE DUAL LEARNING RATE

In Fig 11, we study the impact on the state coverage of METRA in `Ant` for various hyperparameters. Specifically, we vary the initial value of the dual variable $\lambda$, the slack variable $\epsilon$, and the learning rate of the dual variable. We find that METRA is robust to all three hyperparameters. All experiments plot the mean and standard deviation over 2 random seeds.

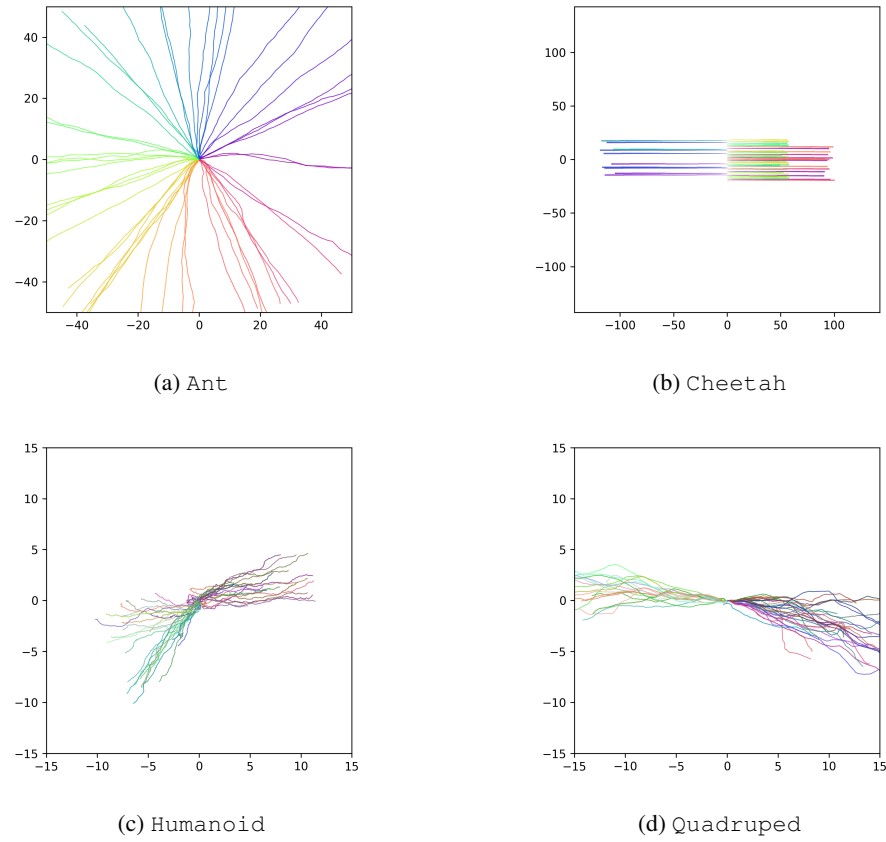

(a) Ant

(b) Cheetah

(c) Humanoid

(d) Quadruped

Figure 12: **Trajectory visualization.** We visualize $(x, y)$ trajectories of different skills (colors) learned by CSF on (a) Ant, (b) Cheetah, (c) Humanoid, and (d) Quadruped, showing that CSF learns diverse locomotion behaviors.

## C.5 SKILL VISUALIZATIONS

In Fig. 12, we visualize $(x, y)$ trajectories of different skills learned by CSF with different colors. We find that CSF learns diverse locomotion behaviors. Videos of learned skills on different tasks can be found on https://princeton-rl.github.io/contrastive-successor-features.

## C.6 FAILED EXPERIMENTS

We experimented with both CSF and METRA on the MiniHack-Corridor-R5-v0 from Mini-Hack (Samvelyan et al., 2021). This environment has a discrete action space and requires the agent to navigate between 5 different rooms using various doors and corridors. None of these methods got consistently stable performance on this environment. We conjecture that the difficulties could come from *(1)* the discrete action space, *(2)* randomized initial states and dynamics (layout of the rooms), and *(3)* partial observability.

