# OpenReview forum: "Can a MISL Fly? Analysis and Ingredients for Mutual Information Skill Learning"
_ICLR.cc/2025/Conference — ICLR 2025 Oral_

### Official Review · Reviewer_eNEc · 2024-10-22

**Soundness:** 4
**Presentation:** 4
**Contribution:** 4
**Rating:** 8
**Confidence:** 3

**Summary:**

The authors introduce a novel self-supervised skill learning algorithm in the RL setting. Their work is motivated by recent work (METRA) that suggests moving away from the typical MI setting, which they analyse, and show that it could be reinterpreted in the familiar MI setting. In doing so, they create a simplified version of METRA, CSF, which achieves the same performance as METRA. The authors combine both theoretical and empirical evidence to support their claims.

**Strengths:**

This paper is incredibly well-written. The message/purpose is clear. There is an extensive literature survey, and the authors' discussion of the relevant material and methodology is informative. Unlike some papers that resort to mathematics unnecessarily, all components seem necessary, and are meaningfully explained in text. The authors perform a relatively large set of experiments to both show the performance of their algorithm, but also to back up other claims (such as the properties of representations learned). I also commend the authors for the extended information in the appendices, and also for providing code (I checked these briefly but not extensively).

**Weaknesses:**

I do not believe there are any substantive weaknesses of this work, but there are some questions the authors could address. As acknowledged by the authors in their own Limitations section, it is unclear how well these algorithms scale beyond the relatively simple MuJoCo benchmarks, but the authors have performed a significant amount of experiments on these domains.

**Questions:**

- Where did the "fixed coefficient ξ = 5" come from/does it have any interpretation? How sensitive is CSR to this hyperparameter?
- There is a note that there are many possible parameterisations for the optimisation of a lower bound on I(S, S′; Z), but CSR (and METRA) use (ϕ(s′)−ϕ(s))⊤z; an ablation study shows that this is a crucial choice. Can the authors provide a further comment on the importance of the temporal distance metric?

---

> ### Author Response · Authors · 2024-11-21
> **Rebuttal by Authors**
>
> We thank the reviewer for the responses and suggestions for improving the paper. The reviewer brought up two main questions about the paper: the interpretation and the effect of the coefficient $\xi$, and the importance of the inner product parameterization for the critic. We have attempted to address these questions by ablating different values of $\xi$ and explaining intuitions of the inner product parameterization respectively. We have also revised the paper to include those interpretations. **Together with the discussion below, does this fully address the reviewer’s questions?**
>
> > Where did the "fixed coefficient ξ = 5" come from/does it have any interpretation?
>
> We introduce the coefficient $\xi$ to scale the negative term $\text{LB}\_{-}^{\beta}(\phi)$ in the contrastive lower bound and found that a proper choice of $\xi$ improves the performance of CSF empirically. This coefficient has an effect similar to the tradeoff coefficient used in information bottleneck optimization ($\beta$ in [1]). We also note that when setting $\xi \geq 1$, the new representation objective $\text{LB}\_{+}^{\beta}(\phi) + \xi \text{LB}\_{-}^{\beta}(\phi)$ is still a (scaled) contrastive lower bound on the mutual information $I^{\beta}(S, S’; Z)$ (see Appendix A.6 for further discussions).
>
> > How sensitive is CSR to this hyperparameter?
>
> To answer this question, we ran ablation experiments to study the effect of the scaling coefficient $\xi$ on the representation objective of CSF empirically. We conducted experiments on the $\texttt{Ant}$ again, selected a set of different values of $\xi \in \\{ 0.5, 1, 2, 5, 10\\}$, and report means and standard deviations of coverage of $(x, y)$ coordinates over 5 random seeds. Results in Fig. 10 (Appendix C.3) suggest that while the state coverage saturates after increasing $\xi$ to $2$, our choice of $\xi = 5$ is optimal for CSF.
>
> > … an ablation study shows that this is a crucial choice. Can the authors provide a further comment on the importance of the temporal distance metric?
>
> Yes. We used inner product parameterization as our distance metric for the critic and this is inherited from METRA. Unlike METRA, our method does not explicitly enforce a $L^2$ temporal distance constraint and instead optimizes the contrastive lower bound on $I^{\beta}(S’, S; Z)$. Our experiments in Fig. 3 (Right) show that the inner product parameterization is important for the good performance of CSF. There are two explanations for these observations. First, prior work [2, 3, 4] has mentioned that mutual information $I(S; Z)$ is invariant to bijective mappings on $S$ (See Fig. 2 of [3] or Fig. 2a of [4]), e.g., translation and scaling, indicating that maximizing the mutual information between *changes of states* and skills ($I(S, S’; Z)$) might encourage better state space coverage. Second, the inner product parameterization allows us to analytically compute the second-order Taylor approximation in Proposition 2, drawing the connection between the METRA representation objective and contrastive learning. We have revised the paper (Appendix A.9 blue texts) to include these insights.
>
> > it is unclear how well these algorithms scale beyond the relatively simple MuJoCo benchmarks
>
> We have already started to investigate applying CSF to MiniHack [5], a simplified version of the immensely complex NetHack [6]. Specifically, we used the MiniHack-Corridor-R5-v0 environment which has a discrete action space and requires the agent to navigate between 5 different rooms using various doors (that require opening) and corridors. We have also implemented a version of METRA for these tasks. Our preliminary results suggest that both METRA and CSF perform similarly after the first 1M env steps, and both explore better than random. Specifically, they both manage to open doors and explore some of the corridors, sometimes even reaching the goal location. We're currently debugging some stability issues for both methods that arise later in training. We'll be sure to include the final results in the final paper.
>
>
> [1] Alexander A. Alemi, Ian Fischer, Joshua V. Dillon, and Kevin Murphy. Deep variational information bottleneck. In International Conference on Learning Representations, 2017.
>
> [2] Park, Seohong, et al. "Lipschitz-constrained unsupervised skill discovery." International Conference on Learning Representations. 2022.
>
> [3] Park, Seohong, Oleh Rybkin, and Sergey Levine. "METRA: Scalable Unsupervised RL with Metric-Aware Abstraction." The Twelfth International Conference on Learning Representations.
>
> [4] Park, Seohong, et al. "Controllability-Aware Unsupervised Skill Discovery." International Conference on Machine Learning. PMLR, 2023.
>
> [5] Samvelyan, Mikayel, et al. "Minihack the planet: A sandbox for open-ended reinforcement learning research." arXiv preprint arXiv:2109.13202 (2021).
>
> [6] Küttler, Heinrich, et al. "The nethack learning environment." Advances in Neural Information Processing Systems 33 (2020): 7671-7684.

---

> > ### Comment · Reviewer_eNEc · 2024-11-22
> > **Response to Authors' Update**
> >
> > Thank you to the authors for their update - I believe these sufficiently address the questions I had. Considering the other reviews and the authors' update, I will be keeping my current recommendation (8).

---

### Official Review · Reviewer_8Sfa · 2024-10-31

**Soundness:** 4
**Presentation:** 4
**Contribution:** 2
**Rating:** 6
**Confidence:** 3

**Summary:**

This work introduces a new approach for learning different locomotor skills without supervision. The paper explains relation between SOTA method (METRA) with mutual information maximization between skills and state transitions, an objective shared by most related works. Following the analysis, it proposes modifications to METRA to make it explicitly maximize such a mutual information. Experiments show that the new approach matches METRA.

Overall, the paper is easy to follow and well-written, the analysis is sound and the experiments are relevant. The main limitation is the novelty of the final approach, which is, in the end, very close to METRA and CIC. It also does not show strong improvements.

I'm balanced but a lean towards borderline reject, I did not find sufficiently strong arguments justifying the proposed approach, compared to METRA (see Questions).

**Strengths:**

The paper is easy to follow and well-written, the analysis is sound and the experiments are relevant.

**Weaknesses:**

The final model is very close to previous work and do not present substantial improvements on the different environment compared to METRA. This is overall acknowledged by the authors.

**Questions:**

The statement line 278 does not hold for all cited methods (like DIAYN), as the "omitted" is sometimes a useless constant. This is actually illustrated by Eq 2 and 3.

I also find the argument that the method removes 5 hyper-parameters compared to METRA to be bad faith: "(1) the ϵ slack variable, (2) the norm constraint value in Eq. 4, (3) the dual gradient descent learning rate, (4) the dual gradient descent optimizer, and (5) the choice of discrete or continuous skills z." The choice between discrete/continuous skills is actually a positive thing, the norm constraint value is not a hyper-parameter, and the Lagrange multiplier seems (as far as I understand) to use the same optimizer/learning rate as the rest of the parameters in the original paper. Could you clarify which hyper-parameter actually require tuning and how many hyper-parameters you introduce ?

---

> ### Author Response · Authors · 2024-11-21
> **Rebuttal by Authors**
>
> We thank the reviewer for the responses and suggestions for improving the paper. It seems like the reviewer’s main question is about the contribution of the paper. Our paper mainly focuses on (1) providing an alternative explanation of METRA theoretically and empirically within the well-studied mutual information skill learning framework, and (2) proposing a simpler algorithm to match the SOTA performance. These contributions are well within scope for ICLR, and ICLR has a history of recognizing papers that make similar contributions [1, 2]. **Together with the discussion below, does this address the reviewer’s concerns?**
>
> > I also find the argument that the method removes 5 hyper-parameters compared to METRA to be bad faith … Could you clarify which hyper-parameter actually require tuning and how many hyper-parameters you introduce?
>
> Thanks for mentioning this; we agree that counting hyperparameters is subjective. For now, we have removed the exact quantification from the main paper and instead said “simpler”; we have also added a new discussion in Appendix B.1 to discuss the hyperparameters that we have removed (and the one hyperparameter $\xi$ that we’ve added) in more detail. We would love to get the reviewer's feedback on this, and would happily consider alternative ways of presenting this comparison.
>
> > The statement line 278 does not hold for all cited methods (like DIAYN), as the "omitted" is sometimes a useless constant.
>
> Thanks for mentioning this. We have revised this sentence to clarify (blue texts). The purpose of this sentence is to contrast that these prior methods use the *same* function form of the lower bound (Eq. 2 $\\&$ 3) on *different* variants of mutual information to learn both representations and policies, while METRA uses different functional form of objectives for the representation and the actor. We elaborate which mutual information each prior method tries to optimize in Appendix A.5 (blue texts).
>
> > The final model is very close to previous work and do not present substantial improvements on the different environment compared to METRA.
>
> Our main contributions are *not* to surpass METRA in terms of performance, but rather the things listed at the top of this response (i.e. explaining METRA within MISL, and proposing a simpler algorithm that matches SOTA).
>
> Nevertheless, to get further insight into the behavior of CSF and METRA, we have already started to investigate applying CSF to MiniHack [3], a simplified version of the immensely complex NetHack [4]. We used the MiniHack-Corridor-R5-v0 environment which has a discrete action space and requires the agent to navigate between 5 different rooms using various doors (that require opening) and corridors. We have also implemented a version of METRA for these tasks. Our preliminary results suggest that both METRA and CSF perform similarly after the first 1M env steps, and both explore better than random. We're currently debugging some stability issues for both methods that arise later in training. We'll be sure to include the final results in the final paper.
>
> [1] Higgins, Irina, et al. "beta-vae: Learning basic visual concepts with a constrained variational framework." International conference on learning representations. 2017.
>
> [2] Arora, Sanjeev, et al. "A simple but tough-to-beat baseline for sentence embeddings." International conference on learning representations. 2017.
>
> [3] Samvelyan, Mikayel, et al. "Minihack the planet: A sandbox for open-ended reinforcement learning research." arXiv preprint arXiv:2109.13202 (2021).
>
> [4] Küttler, Heinrich, et al. "The nethack learning environment." Advances in Neural Information Processing Systems 33 (2020): 7671-7684.

---

> > ### Comment · Reviewer_8Sfa · 2024-11-21
> > **Response**
> >
> > I thank the authors for their clear answer and the modifications of the paper. I acknowledge that the paper makes a significant analysis of METRA, thus I will increase my score.
> >
> > > Thanks for mentioning this; we agree that counting hyperparameters is subjective. For now, we have removed the exact quantification from the main paper and instead said “simpler”; we have also added a new discussion in Appendix B.1 to discuss the hyperparameters that we have removed (and the one hyperparameter that we’ve added) in more detail. We would love to get the reviewer's feedback on this, and would happily consider alternative ways of presenting this comparison.
> >
> > An analysis of the sensitivity of the method with respect to $\lambda$, $\epsilon$ and the learning rate of the dual variable optimizer (versus the coefficient that scales the negative term, which is already provided in Appendix) may also help to appreciate the simplicity of CSF in terms of hyperparameters. Removing "almost constant" hyperparameters does not really make a method simpler.  Showing the algorithms of METRA and CSF side by side may also support that CSF is "simpler to implement" (l196).

---

> ### Author Response · Authors · 2024-11-22
> **Author Response**
>
> Thanks again for your review and further suggestions. We are running ablation experiments studying the sensitivity of METRA with respect to $\lambda$, $\epsilon$, and the learning rate of the dual variable optimizer, helping to understand the simplicity of CSF versus METRA. We will be sure to include discussions about these in the final version.

---

### Official Review · Reviewer_PYdS · 2024-11-02

**Soundness:** 3
**Presentation:** 3
**Contribution:** 3
**Rating:** 8
**Confidence:** 3

**Summary:**

This work makes two major contributions. First, it establishes an approximate equivalence that bridges the representation objectives of the state-of-the-art method METRA with contrastive loss, specifically similar to InfoNCE. It shows that the actor objective in METRA is equivalent to the information bottleneck of $I(S, S'; Z) - I(S, S'; \phi(S') - \phi(S))$ (lower bounded). Essentially, it uses mutual-information-based skill discovery to elucidate METRA. Building on the analytical framework, the authors propose contrastive successor features to simplify METRA, employing contrastive objectives for representation learning and successor features for policy learning. Results indicate that the proposed method is empirically competitive with METRA.

Overall, I like the analytical framework that unifies METRA with the mutual-information-based skill discovery method, and the theoretical foundation appears solid. Additionally, the results support the hypotheses and propositions, suggesting that the proposed method is even more flexible than METRA. Given these strengths, I would recommend an accept in this initial review.

**Strengths:**

- **[Technical soundness and novelty]** The technical soundness is robust; this work provides a thorough in-depth analysis of the METRA method, finding approximate equivalences with contrastive objectives and the information bottleneck. The analysis leads to a novel method that simplifies METRA, and I found no technical flaws; the method is both novel and solid.


- **[Evaluation]** The empirical evaluation effectively validates the hypotheses and theoretical analysis, enhancing the overall persuasiveness of the work.

- **[Presentation]** The presentation is clear and easy to follow.

**Weaknesses:**

- **[About performances]** A significant question arises especially in the Quadruped experiments, where performance still shows room for improvement compared to METRA. Given that the proposed framework has a similar objective function, fewer hyperparameters, and avoids complex min-max optimization, why does the empirical performance (or at least the rate of convergence) not exceed that of METRA? Any discussion on this would be beneficial.
- **[About demonstrations]** It would be advantageous to include demonstrations or other forms of visualization for the skills learned, as I did not find this in the appendix code.

**Questions:**

- Are there any theoretical insights supporting the assertion that parameterization is key for contrastive successor features? Did you experiment with different kernel functions?

- Any insights or rough ideas on how to further scale the framework to more complicated cases, perhaps through large-scale pre-training or using foundation models to replace the representation learning component, would be beneficial to include in the main paper

---

> ### Author Response · Authors · 2024-11-21
> **Rebuttal by Authors - Part 1**
>
> We thank the reviewer for the response, and for the suggestions for improving the paper. To address the concerns about experiments, we have run additional experiments to ablate different distance metrics / kernel functions for the critic. **Together with the discussion for other questions below, does this fully address the reviewer’s concerns about the paper?**
>
> >  Are there any theoretical insights supporting the assertion that parameterization is key for contrastive successor features?
>
> Yes. Our experiments show that the inner product parameterization of the critic $f(s, s’, z) = (\phi(s’) - \phi(s))^{\top} z$ is important for CSF (Fig. 3 (Right)). There are two explanations for these observations.
>
> First, the inner product parameterization tries to push together the *difference* of the representation of transitions $\phi(s’) - \phi(s)$ and the corresponding skill $z$, instead of focusing on representations of individual states $\phi(s)$ or $\phi(s’)$. This intuition is inline with the observation that mutual information $I(S; Z)$ is invariant to bijective mappings on $S$ (See Fig. 2 of [1]), e.g., translation and scaling, indicating that maximizing the mutual information between *change of states* and skills ($I(S, S’; Z)$) encourages better state space coverage.
>
> Second, the inner product parameterization allows us to analytically compute the second-order Taylor approximation in Proposition 2, drawing the connection between the METRA representation objective and contrastive learning. We have revised the paper to include these theoretical insights (see Appendix A.9 blue texts).
>
> > Did you experiment with different kernel functions?
>
> As suggested by the reviewer, we ran additional experiments to study the effect of different distance metrics / kernel functions for the critic in Fig. 3 (Right). We selected $L^1$ distance (Laplacian kernel) $-\\| \phi(s') - \phi(s) - z \\|_1$ and squared $L^2$ distance  (Gaussian kernel) $-\frac{1}{2} \left \\| \phi(s') - \phi(s) - z \right \\|_2^2$ in addition to the inner product (vMF kernel) and the monolithic MLP parameterizations of the critic to conduct ablation studies. For the sake of consistency, we chose to use the $\texttt{Ant}$ task and report means and standard deviation of coverage of unique (x, y) coordinates over 5 random seeds. Results in Fig. 3 (Right) suggest that the inner product critic is the best for CSF among different choices of parameterizations.
>
> > It would be advantageous to include demonstrations or other forms of visualization for the skills learned, as I did not find this in the appendix code.
>
> Thanks for the suggestion. We visualized skills learned by CSF and METRA (anonymous videos in https://anonymous.4open.science/r/csf-3BF4/README.md under “Videos of learned policies” section) and have revised the paper to include visualizations of $(x, y)$ trajectories of these skills in Appendix C.4 (blue texts).
>
> > A significant question arises especially in the Quadruped experiments … why does the empirical performance (or at least the rate of convergence) not exceed that of METRA?
>
> Our hypothesis for this observation is that METRA might overfit to some of the tasks. The reason for this is that METRA uses dual gradient descent to optimize the representation objective, resulting in 2 more hyperparameters than CSF (see Appendix B.1). While using more hyperparameters benefits the flexibility of METRA [2], it also means that the algorithm tends to overfit especially when the effective dimensions of a task are low, e.g., $(x, y)$ in Quadruped.
>
> We note that CSF only slightly underperforms METRA (within one standard deviation confidence interval) in terms of both the average state coverage and the rate of convergence. As supported by our theory, the main practical benefit of CSF is *not* that it should achieve better results on every task but rather that it is easier to implement (see algorithm summary in Sec. 5.2 and Appendix B.1).

---

> > ### Author Response · Authors · 2024-11-21
> > **Rebuttal by Authors - Part 2**
> >
> > > Any insights or rough ideas on how to further scale the framework to more complicated cases, perhaps through large-scale pre-training or using foundation models to replace the representation learning component …
> >
> > As suggested by the reviewer, large-scale pre-training would be a natural next step of our work. Some important things to keep in mind would be:
> >
> > 1. **Complex environments**: environments like Avalon [3], Minecraft [4], NetHack [5], Isaac Gym [6], or VIMA [7] would allow us to investigate whether our method would keep learning new skills or plateau at some point.
> >
> > 2. **Fast / gpu-accelerated environments**: scale has resulted in dramatic performance in NLP, so it seems plausible that it might play a crucial role here as well when thinking about training foundation models that learning increasingly complex skills.
> >
> > 3. **Good evaluation metrics**: as the environments and learned behaviors become more complex, we’ll need good evaluation metrics in order to track what kind of skills are learned.
> >
> > We have revised the limitation section to include some discussions.
> >
> > [1] Park, Seohong, Oleh Rybkin, and Sergey Levine. "METRA: Scalable Unsupervised RL with Metric-Aware Abstraction." The Twelfth International Conference on Learning Representations.
> >
> > [2] Recht, Benjamin, et al. "Do imagenet classifiers generalize to imagenet?." International conference on machine learning. PMLR, 2019.
> >
> > [3] Albrecht, Joshua, et al. "Avalon: A benchmark for RL generalization using procedurally generated worlds." Advances in Neural Information Processing Systems 35 (2022): 12813-12825.
> >
> > [4] Fan, Linxi, et al. "Minedojo: Building open-ended embodied agents with internet-scale knowledge." Advances in Neural Information Processing Systems 35 (2022): 18343-18362.
> >
> > [5] Küttler, Heinrich, et al. "The nethack learning environment." Advances in Neural Information Processing Systems 33 (2020): 7671-7684.
> >
> > [6] Makoviychuk, Viktor, et al. "Isaac gym: High performance gpu-based physics simulation for robot learning." arXiv preprint arXiv:2108.10470 (2021).
> >
> > [7] Jiang, Yunfan, et al. "Vima: General robot manipulation with multimodal prompts." arXiv preprint arXiv:2210.03094 2.3 (2022): 6.

---

> > > ### Comment · Reviewer_PYdS · 2024-11-22
> > > **Response to the rebuttal**
> > >
> > > Thank you for the detailed response. The discussion together with the revised parts addressed my concerns. I would keep my rating (8).

---

### Official Review · Reviewer_8yQh · 2024-11-02

**Soundness:** 4
**Presentation:** 4
**Contribution:** 4
**Rating:** 8
**Confidence:** 2

**Summary:**

The paper critiques a recent method (METRA), which optimizes a Wasserstein distance for skill learning, and argues that its benefits can be explained within the existing framework of mutual information skill learning (MISL). The authors propose a new MISL method called Contrastive Successor Features (CSF), which retains METRA's performance with fewer complexities (namely fewer hyperparameters but same performance).
The paper highlights connections between skill learning, contrastive representation learning, and successor features, and provides insights through ablation studies.

**Strengths:**

The paper provides a thorough theoretical analysis of METRA, reinterpreting it within the mutual information skill learning (MISL) framework. This helps demystify the method and connects it to well-established concepts like contrastive learning and information bottlenecks.

The presentation is clear and to the point. The writing is excellent, I did not find typos or mistakes.

The paper includes extensive empirical evaluations, comparing CSF with existing methods across various tasks. This robust experimental setup strengthens the validity of the proposed method.

**Weaknesses:**

While I appreciate that the paper is presented as an improvement on METRA, I'd have enjoyed more a reading that was presenting a new method that is then shown to be equivalent to METRA under certain conditions.

Given that the presented method performs are par with METRA, it would also be nice to show where (if anywhere) one fails when the other succeeds. Perhaps partially observed MPDs, more interactive objects or discrete actions spaces would be key in identifying where exactly both methods stand.

**Questions:**

- Can the authors elaborate on the assumptions made in the theoretical analysis, and how they might affect the generalizability of the results? It is not clear from the main text if these assumptions are sound and/or restrictive in any significant way.

---

> ### Author Response · Authors · 2024-11-21
> **Rebuttal by Authors**
>
> We thank the reviewer for the responses and helpful suggestions for improving the paper. The reviewer mainly raised two questions about the paper: evaluation on diverse benchmarks and conditions under which the theoretical analysis holds. We have attempted to address the question about evaluations by starting to run experiments on MiniHack benchmarks [1], and we answer the question about conditions and assumptions for theoretical analysis below. **Together with the discussion below, does this address the reviewer’s concerns?**
>
> > … it would also be nice to show where (if anywhere) one fails when the other succeeds. Perhaps partially observed MPDs, more interactive objects or discrete actions spaces would be key in identifying where exactly both methods stand.
>
> We have included the full results of the state space coverage in Appendix B.3 and Fig. 5 shows that CSF matches the prior SOTA performance on 4 / 6 tasks. Specifically, CSF outperforms METRA on the $\texttt{Robobin (Pixels)}$ while slightly underperforming METRA on the $\texttt{Quadruped (Pixels)}$.
>
> In addition, as suggested by the reviewer, we have started to investigate applying CSF to MiniHack [1], a simplified version of the immensely complex NetHack [2]. Specifically, we used the MiniHack-Corridor-R5-v0 environment which has a discrete action space and requires the agent to navigate between 5 different rooms using various doors (that require opening) and corridors. We have also implemented a version of METRA for these tasks. Our very preliminary results suggest that both METRA and CSF perform similarly after the first 1M gradient steps, and both explore better than random. We're currently debugging some stability issues for both methods that arise later in training. We'll be sure to include the final results in the final paper.
>
> > Can the authors elaborate on the assumptions made in the theoretical analysis, and how they might affect the generalizability of the results?
>
> Our theoretical analysis mainly builds on four conditions (assumptions) which are inline with empirical findings in the didactic experiments. Below, we discuss these assumptions in detail.
>
> First, when casting the mutual information maximization problem into a min-max optimization problem, we assume the realizability of the variational function class $Q = \\{q(z \mid s, s’)\\}$, i.e., $Q$ is expressive enough to cover the ground truth posterior $p^{\pi}(z \mid s, s’)$. Many prior MISL methods [3, 4, 5] adopt the min-max optimization procedure and implicitly apply this assumption.
>
> Second, our analysis focuses on the case where $p(z)$ is distributed as a uniform distribution on the $d$-dimensional unit hypersphere, as is common in much (but not all) prior work [3, 6]. Using this prior distribution helps us to unify analysis for both the representation objective and the policy objective in METRA.
>
> Third, in Proposition 2, we claim that the constraints used in the METRA representation objective correspond to a second order Taylor approximation of the negative term in the contrastive lower bound ($\text{LB}_{-}^{\beta}(\phi)$). However, since METRA uses dual gradient descent to adaptively update the dual variable $\lambda$, the tightness of the Taylor approximation depends on the value of $\lambda$. We implicitly assume that the Taylor approximation holds and the METRA representation objective has the same effect as contrastive learning. In practice, our experiments in Fig. 8 and Appendix C.1 verify that this assumption is reasonable. Note that we also apply this Taylor approximation in the proof of Proposition 3 (step (c)).
>
> Finally, we apply Proposition 1 in the proof of Proposition 3 (step (d)), which implicitly assumes that the optimization problem in Eq. 7 is solved without errors. We also empirically verify this condition in the didactic experiments Fig. 2a and Sec. 6.1.
> We have revised the paper accordingly (Sec. 3 and Appendix A.4 blue texts) to clarify these conditions.
>
> [1] Samvelyan, Mikayel, et al. "Minihack the planet: A sandbox for open-ended reinforcement learning research." arXiv preprint arXiv:2109.13202 (2021).
>
> [2] Küttler, Heinrich, et al. "The nethack learning environment." Advances in Neural Information Processing Systems 33 (2020): 7671-7684.
>
> [3] Eysenbach, Benjamin, et al. "Diversity is All You Need: Learning Skills without a Reward Function." International Conference on Learning Representations.
>
> [4] Hansen, Steven, et al. "Fast Task Inference with Variational Intrinsic Successor Features." International Conference on Learning Representations.
>
> [5] Gregor, Karol, Danilo Jimenez Rezende, and Daan Wierstra. "Variational intrinsic control." arXiv preprint arXiv:1611.07507 (2016).
>
> [6] Park, Seohong, Oleh Rybkin, and Sergey Levine. "METRA: Scalable Unsupervised RL with Metric-Aware Abstraction." The Twelfth International Conference on Learning Representations.

---

> > ### Comment · Reviewer_8yQh · 2024-11-25
> > **Final assessment**
> >
> > I would like to thank the authors for thoroughly addressing my concerns in their response.
> > The final version would definitely benefit from more experiments if you can add them!
> >
> > I remain impressed with the work, I am confident that the contributions presented are meaningful and valuable to the field. I will keep my rating of 8 unchanged, primarily because of the limitations of my own expertise in this area.

---

> ### Comment · Area_Chair_YDnv · 2024-11-23
> **From AC.**
>
> Reviewer 8yQh: if you can, please reply the rebuttal.

---

### Meta-Review · Area_Chair_YDnv · 2024-12-19

**Metareview:**

The paper discusses unsupervised learning of skills. The paper proposes a new method (contrastive successor features), which works within the paradigm of mutual information skill learning. The new method matches the performance of METRA, a method based on optimal transport, which achieves state of the art performance.

The main strengths of the paper are: (1) in-depth analysis of METRA, (2) clear writing, (3) careful ablations.

The main weakness is that empirical performance doesn't exceed METRA (this is fine given the framing of the paper).

Overall, because of the stregths outlined above, I recommend acceptance.

**Additional Comments On Reviewer Discussion:**

All reviewers agree this should be accepted.

While I am recommending a spotlight, I wouldn't mind if this gets bumped to an oral.

---

### Decision · Program_Chairs · 2025-01-22

Accept (Oral)